

# Soil Methanotrophy Model (MeMo v1.0): a process-based model to quantify global uptake of atmospheric methane by soil

Fabiola Murguia-Flores[1], Sandra Arndt[1,2], Anita L. Ganesan[1], Guillermo N. Murray-Tortarolo[3,4], Edward R.C. Hornibrook[5,6]

[1]School of Geographical Sciences, University of Bristol, Bristol, BS8 1SS, United Kingdom
[2]*Current address:* Department of Geosciences, Environment and Society, Université Libre de Bruxelles, Brussels, Belgium
[3]University of Exeter, Devon, EX4, United Kingdom
[4] *Current address:* Instituto de Investigaciones en Ecosistemas y Sustentabilidad (IIES), UNAM.
[5]School of Earth Sciences, University of Bristol, Bristol, BS8 1RJ. United Kingdom
[6]*Current address:* Earth, Environmental and Geographic Sciences, The University of British Columbia, Okanagan Campus, Kelowna, BC, Canada V4V 1C7

*Correspondence to*: Fabiola Murguia-Flores (fmurguia84@gmail.com)

**Abstract.**

**Soil bacteria known as methanotrophs are the sole biological sink for atmospheric methane ($CH_4$), a powerful greenhouse gas that is responsible for ~20% of the human-driven increase in radiative forcing since pre-industrial times. Soil methanotrophy is controlled by a plethora of different factors, including temperature, soil texture and moisture or nitrogen content, resulting in spatially and temporally heterogeneous rates of soil methanotrophy. As a**
**consequence, the exact magnitude of the global soil sink, as well as its temporal and spatial variability remains poorly constrained. We developed a process-based model (Methanotrophy Model; MeMo v1.0) to simulate and quantify the uptake of atmospheric $CH_4$ by soils on the global scale. MeMo builds on previous models by Ridgwell et al. (1999) and Curry (2007) by introducing several advances, including: (1) a general analytical solution of the one-dimensional diffusion-reaction equation in porous media, (2) a refined representation of nitrogen inhibition on soil**
**methanotrophy, and (3) updated factors governing the influence of soil moisture and temperature on $CH_4$ oxidation rates. We show that the improved representation of these key drivers of soil methanotrophy resulted in a better fit to observational data. A global simulation of soil methanotrophy for the period 1990-2009 using MeMo yielded an average annual sink of $34.3 \pm 4.3$ Tg $CH_4$ yr$^{-1}$. Warm and semiarid regions (tropical deciduous forest, dense and open shrubland) had the highest $CH_4$ uptake rates of 630 and 580 mg $CH_4$ m$^{-2}$ y$^{-1}$, respectively. In these regions, favorable**
**annual soil moisture content (~20% saturation) and low seasonal temperature variations (variations < ~6 ºC) provided optimal conditions for soil methanotrophy and soil-atmosphere gas exchange. In contrast to previous model**



analyses, but in agreement with recent observational data, MeMo predicted low fluxes in wet tropical regions because of refinements in describing the influence of excess soil moisture on methanotrophy. Tundra and boreal forest had the lowest simulated $CH_4$ uptake rates of 179 and 187 mg $CH_4$ $m^{-2}$ $y^{-1}$, respectively, due to their marked seasonality driven by temperature. Soil uptake of atmospheric $CH_4$ was attenuated by up to 10% in regions receiving high rates of nitrogen deposition. Globally, nitrogen deposition reduced soil uptake of atmospheric $CH_4$ by 0.34 Tg $y^{-1}$, which is an order of magnitude lower than reported previously. In addition to improved characterization of the contemporary soil sink for atmospheric $CH_4$, MeMo provides an opportunity to quantify more accurately the relative importance of soil methanotrophy in the global $CH_4$ cycle in the past and its capacity to contribute to reduction of atmospheric $CH_4$ levels under future global change scenarios.

## 1 Introduction

Methane ($CH_4$) is the most abundant organic trace gas in the atmosphere and responsible for approximately 20% of the human-driven increase in radiative forcing since preindustrial times (Myhre et al., 1998; Ciais et al., 2013). Anthropogenic activities during the last 200 years have increased the concentration of $CH_4$ in the atmosphere from pre-industrial levels of approximately 710 parts per billion (ppb) to the current mixing ratio of approximately 1800 ppb (Etheridge et al., 1998; Kirschke et al., 2013). The atmospheric lifetime of $CH_4$ is $9.1 \pm 0.9$ years (Prather et al., 2012) and most $CH_4$ is consumed in the troposphere via oxidation by OH radicals, which represents ~90% of the global $CH_4$ sink (Prather et al., 2012; Ciais et al., 2013). Soil bacteria known as methanotrophs consume ~9 to 10% of atmospheric $CH_4$ and a further ~1% is oxidized by reaction with chlorine radicals from sea salt in the marine boundary layer (Allan et al., 2007; Ciais et al., 2013).

Soil methanotrophy is the only biological sink for $CH_4$ and its rate is highly dependent on environmental conditions. The total global soil sink is similar in size to global emissions of $CH_4$ from rice paddies (Kirschke et al., 2013) and consequently year-to-year changes in factors that impact rates of soil $CH_4$ oxidation may contribute to variability in the interannual growth rate of atmospheric $CH_4$. Moreover, soil methanotrophy consumes up to 90% of $CH_4$ produced via methanogenesis in persistently or periodically wet soil and thus factors that impact soil uptake of atmospheric $CH_4$ may reduce the capacity of soil methanotrophs to attenuate emission of soil-produced $CH_4$ (Oremland and Culbertson, 1992; Singh et al., 2010).

The rate of methanotrophy in soil is controlled by several environmental factors including temperature, soil texture, moisture and nitrogen (N) content (Czepiel et al., 1995; Le Mer and Roger, 2001; Wang et al., 2005). The influence of these factors on rates of $CH_4$ oxidation has been widely studied both at the ecosystem level and under laboratory conditions. Positive correlations have been consistently reported between temperature and rates of $CH_4$ oxidation in soil (Castro et al., 1995; Butterbach-Bahl and Papen, 2002; Rosenkranz et al., 2006; Luo et al., 2013). Atypically low and high soil moisture levels both have a negative impact on rates of atmospheric $CH_4$ consumption. A soil moisture content of ~20% appears to



yield optimum rates of CH$_4$ uptake in different ecosystems, including tropical forests, short grass steppe and tundra (Adamsen and King, 1993; Mosier, 2002; Burke et al., 1999; Castro et al., 1995; Epstein et al., 1998; Klemedtsson and Klemedtsson, 1997; McLain and Ahmann, 2007; West et al., 1999). Soil texture impacts the ability of soil to retain water and influences diffusion of atmospheric CH$_4$ and O$_2$ into soil because of its control on pore size and connectivity. Thus

sandy soil generally exhibits higher rates of CH$_4$ uptake than silt-rich soil followed by clayey soil (Born et al., 1990; Dörr et al., 1993). The influence of N input from atmospheric deposition and fertilizer application is more complex; however, the majority of studies report inhibition of soil methanotrophy with increased addition of N (Aronson and Helliker, 2010; Bodelier and Laanbroek, 2004; Fang et al., 2014).

There is a large year-to-year uncertainty in the accounting of the global CH$_4$ budget, particularly for processes that

consume CH$_4$ (Kirschke et al., 2013). Our understanding of the main drivers of CH$_4$ uptake in soils and how those factors respond to climate change is incomplete. Estimates of the soil CH$_4$ sink based upon field data (Dutaur and Verchot, 2007) show high variability globally and within different ecosystems. Numerical models provide an efficient means to deal with the spatial and temporal heterogeneity and to evaluate mechanistic understanding of physical and biological processes that influence soil methanotrophy. Ultimately, models enable derivation of regional and global estimates of soil uptake of

atmospheric CH$_4$ and provide the ability to predict the response of soil methanotrophy to past and future global change. In addition, they provide a platform of interdisciplinary knowledge synthesis, help identify the most important parameters and environmental controls and can thus inform future field and laboratory research.

Several biogeochemical models have been developed to quantify global consumption of atmospheric CH$_4$ by soil. The model of Potter et al. (1996) (hereafter referred to as 'P96' model) estimated terrestrial uptake of CH$_4$ by calculating

diffusive flux of atmospheric CH$_4$ into soil using a modified version of Fick´s first law. Ridgwell et al. (1999) (hereafter referred to as 'R99' model) improved the P96 model by explicitly accounting for microbial CH$_4$ oxidation in soil. The R99 model quantifies CH$_4$ oxidation rates as a function of soil temperature, moisture and N content. The latter parameter was estimated using agricultural land area as a proxy for fertilizer application. Solution of the resulting one-dimensional diffusion-reaction equation was approximated semi-numerically assuming steady state conditions. Curry (2007) (hereafter

referred to as 'C07'model) employed a steady state analytical solution of the one-dimensional diffusion-reaction equation and introduced a scalar modifier to account for the regulation of CH$_4$ oxidation rates by soil moisture and the impact of temperature below 0°C. The C07 model continued to use the R99 agricultural land area approximation to evaluate the effect of N loading on CH$_4$ uptake. The C07 model has been employed as a reference model for the Global Carbon Project (Kirschke et al., 2013) and has been used to estimate global CH$_4$ uptake in dynamic global vegetation models, such as the

Lund-Potsdam-Jena model (LPJ-WHy-Me; Wania et al., 2010; Spahni et al., 2011).

Over the past decade, the increased availability of new observational and experimental data has led to an improved characterisation of CH$_4$ oxidation processes in soil and thus, an opportunity to revise and enhance model parameterization and validation. For example, new global datasets quantifying N deposition now enable better representation of this key inhibitory effect on soil uptake of atmospheric CH$_4$ (Lamarque et al., 2013).





Here we present an updated process-based model to quantity the global sink for atmospheric $CH_4$ by soil (hereafter referred to as 'MeMo': soil **Me**thanotrophy **Mo**del). MeMo builds on the R99 and C07 models; however, it is based on a more general analytical solution of the one-dimensional diffusion-reaction equation, which makes obsolete the *a priori* assumption of complete $CH_4$ consumption in the model domain applied in the C07 model. The refinement now also

provides the opportunity to account for $CH_4$ flux from below (*i.e.*, due to $CH_4$ production in soil, if present). In addition, MeMo revisits and improves R99 and C07 model formulations to incorporate advances in the mechanistic understanding of soil methanotrophy that have resulted from availability of new data. Finally, MeMo utilizes for the first time N deposition data to explore more accurately the effect of land-use and land-use changes on the global $CH_4$ sink. We present a comprehensive description of the new model, a comparison of MeMo with the R99 and C07 models, and a critical discussion

of model formulations and assumptions based on observational data. We then provide an assessment of global and regional soil uptake and variability across ecosystem types and seasons.

## 2.0 Model Description

The following sections provide a detailed description of MeMo in the context of existing global soil $CH_4$ uptake. Table 1 provides a summary of all terms, names and units used in the model description section, while Table 2 contains a short

summary of the four existing global $CH_4$ uptake models.

**Table 1. Terms, names and units used in the model description section.**

| Terms | Name | Units |
|---|---|---|
| $CH_4$ | $CH_4$ concentration | mg m$^{-3}$ |
| $J_{CH4}$ | $CH_4$ flux uptake | mg $CH_4$ m$^{-2}$ mo$^{-1}$ |
| $C_{CH4}$ | Atmospheric $CH_4$ concentration | ppb |
| $F_{CH4}$ | $CH_4$ flux through $L$ | mg $CH_4$ m$^{-2}$ mo$^{-1}$ |
| $A$ and $B$ | Integration constants | dimensionless |
| $z$ | Depth in the soil profile | cm |
| $L$ | Depth of 99.9% penetration of atmospheric $CH_4$ into the soil | cm |
| $D_{CH4}$ | Diffusion coefficient of $CH_4$ into soil | cm$^2$ s$^{-1}$ |
| $k_d$ | $CH_4$ oxidation activity | s$^{-1}$ |
| $D_{0CH4} = 0.196$ | $CH_4$ diffusion in free air at standard temperature and pressure STP= 0°C and 1 atm pressure | cm$^2$ s$^{-1}$ |
| $G_T$ | Soil temperature response | °C |
| $G_{soil}$ | Soil structure response | dimensionless |
| $\Phi$ | Total pore volume | cm$^3$ cm$^{-3}$ |
| $\rho$ | Bulk density | cm$^{-3}$ g$^{-1}$ |
| $\mathrm{d} = 2.65$ | Soil particle density | g cm$^{-3}$ |
| $\Phi_{air}$ | Air-filled porosity | cm$^3$ cm$^{-3}$ |
| $\theta$ | Soil water content | % |
| $w$ | Saturation soil water potential | MPa |
| $b$ | Clay soil content factor | dimensionless |
| $f_{clay}$ | Clay soil content | % |
| $k_0$ | Base oxidation rate constant for uncultivated moist soil at 0°C | s$^{-1}$ |



| $r_{SM}$ | Microbial $CH_4$ oxidation, soil moisture response | dimensionless |
| $r_T$ | Microbial $CH_4$ oxidation, temperature response | dimensionless |
| $r_N$ | Microbial $CH_4$ oxidation, nitrogen response | dimensionless |
| $N_{soil}$ | Nitrogen deposition into soil | g N m$^{-2}$ mo$^{-1}$ |
| $\alpha = 0.33$ | Average coefficient of N deposition inhibition | % mol N$^{-1}$ |

## 2.1 Conservation Equation

The general, one-dimensional mass conservation equation for $CH_4$ in soil is given by:

$$\frac{\partial CH_4}{\partial t} = -\frac{\partial J_{CH4}}{\partial z} + \sum R \tag{1}$$

Where $J_{CH4}$ denotes the flux of $CH_4$ and $\Sigma R$ is the sum of all production and consumption processes that affect $CH_4$ concentrations in soil. The flux $J_{CH4}$ in the soil is generally controlled by diffusion. Consequently, the P96 model assumes

10  that global uptake of atmospheric $CH_4$ by soil is diffusion limited and thus describes the soil $CH_4$ sink as a purely diffusive process (*i.e.*, $\sum R = 0$). However, $CH_4$ is consumed by microbial activity in the soil and the simplified diffusion model may thus underestimate total uptake of $CH_4$. Consequently, R99 extended the diffusion model by explicitly accounting for microbial oxidation of $CH_4$ through a first order rate expression. The resulting diffusion- reaction equation forms the basis of the R99 model, the C07 model and MeMo:

$$\frac{\partial CH4}{\partial t} = -D_{CH4}\,\frac{\partial^2 CH4}{\partial z^2} + k_d * CH_4 \tag{2}$$

Where $D_{CH4}$ is the $CH_4$ diffusion coefficient and $k_d$ the first-order rate constant for microbial $CH_4$ oxidation. Under steady-state conditions (*i.e.*, $\partial CH_4/\partial t = 0$), soil $CH_4$ uptake is controlled by the balance between diffusion of $CH_4$ into soil

20  and the rate of microbial $CH_4$ oxidation. Hence, accurate characterization of $D_{CH4}$ and $k_d$ is essential for a robust quantification of $CH_4$ uptake by soil.

## 2.2 Solution of Reaction-Transport Equation

The R99 model solved Eq. (2) semi-numerically by (i) assuming steady-state, (ii) numerically approximating the diffusion

25  term similar to the approach applied in the P96 model (Table 2, Eq. 11), and (iii) assigning $CH_4$ oxidation exclusively to a distinct soil layer of thickness $\epsilon$ at depth $z_d = 6$ cm (Table 2, Eq. 12). However, $CH_4$ consumption can occur throughout a soil profile and thus Eq. (12) (Table 2) may either overestimate or underestimate the $CH_4$ sink.



In the C07 model, Eq. (2) was solved analytically, providing a more accurate and mathematically robust estimate of CH$_4$ uptake Eq. (13) (Table 2). Assuming steady-state and constant $D_{CH4}$ and $k_d$ throughout the soil profile, integration of Eq. (2) provides a general solution for determining CH$_4$ concentration at depth $z$ in soil:

$$CH_4(z) = A * exp\left(-\sqrt{\frac{k_d}{D_{CH4}}}z\right) + B\,exp\left(\sqrt{\frac{k_d}{D_{CH4}}}z\right) \tag{3}$$

Where $A$ and $B$ are integration constants that can be determined by setting upper and lower boundary conditions for the soil profile. The concentration of CH$_4$ at the soil-atmosphere interface is defined by the atmospheric concentration of CH$_4$ ($C_{CH4}$) and thus, a Dirichlet boundary (*i.e.*, fixed concentration) is applied at the upper boundary. Conditions at the lower boundary are more challenging to ascribe because the soil depth at which atmospheric CH$_4$ is completely consumed is not known *a priori*.

### *Negligible CH$_4$ flux through the lower boundary (C07 Solution)*

The C07 model circumvents the problem by applying a homogenous Neumann (no-flux) condition at the lower model boundary: $\frac{dCH_4}{d_z}|_{z->\infty} = 0$

The application of this boundary condition allows derivation of the integration constants $A = C_{CH4}$ and $B = 0$, which simplifies Eq. (3) to:

$$CH_4(z) = C_{CH4} * exp\left(-\sqrt{\frac{k_d}{D_{CH4}}} * z\right) \tag{4}$$

The diffusive uptake of atmospheric CH$_4$ at the soil-atmosphere interface can then be calculated using the derivative of Eq. (4) at $z = 0$:

$$J_{CH4} = -D_{CH4} * \frac{dCH4}{d_z}|_{z=0} = D_{CH4} * C_{CH4} * \sqrt{\frac{k_d}{D_{CH4}}} = C_{CH4}\sqrt{D_{CH4}k_d} \tag{5}$$

This formulation of soil uptake is the simplest analytical solution to Eq. (2). It represents an improvement from the semi-numerical representation used in the R99 model and enables complete consumption of CH$_4$ to be accounted for within the soil layer; however, the homogeneous Neumann boundary condition applied here is only an approximation, which is not generally valid. The simulation will not be influenced if the Neumann boundary is infinitely far from the consumption depth of CH$_4$ and thus, the corresponding Neumann boundary conditions can be neglected. However, if this is not the case, it will result in simulation error.




*Complete consumption of CH₄ at an a priori unknown depth L (MeMo solution)*

Therefore, we adopted an approach similar to the C07 model but one that is generally valid. We assume that methanotrophy consumes atmospheric $CH_4$ in the soil until $CH_4$ is fully depleted at an *a priori* unknown depth $L$ (i.e. $CH_4$ ($L$) = 0). The integration constants in Eq. (3) thus become:

$$A = \frac{c_{CH4} * exp\left(\sqrt{\frac{k_d}{D_{CH4}}}L\right)}{\left[-exp\left(-\sqrt{\frac{k_d}{D_{CH4}}}L\right) + exp\left(\sqrt{\frac{k_d}{D_{CH4}}}L\right)\right]} \tag{6}$$

$$B = \frac{c_{CH4} * exp\left(-\sqrt{\frac{k_d}{D_{CH4}}}L\right)}{\left[-exp\left(-\sqrt{\frac{k_d}{D_{CH4}}}L\right) + exp\left(\sqrt{\frac{k_d}{D_{CH4}}}L\right)\right]} \tag{7}$$

In addition to the concentration condition $CH_4$ ($L$)= 0, a flux condition also is imposed on the lower boundary in order to determine depth $L$: $-D_{CH4} * \frac{dCH4}{d_z}|_{z=L} = F_{CH4}$

Where $F_{CH4}$ denotes a potential $CH_4$ flux across the lower boundary that can be specified or set equal to zero. The unknown depth $L$ at which $CH_4 = 0$ is then calculated by substituting the derivative of Eq. (3) into the expression for the lower boundary condition:

$$-D_{CH4} * \frac{dCH4}{dz}|L = -D_{CH4} * \left(A\left(-\sqrt{\frac{k_d}{D_{CH4}}}\right) * exp\left(-\sqrt{\frac{k_d}{D_{CH4}}}L\right) + B\sqrt{\frac{k_d}{D_{CH4}}} * exp\left(\sqrt{\frac{k_d}{D_{CH4}}}L\right)\right) = F_{CH4} \tag{8}$$

Rearranging Eq. (8) and finding its root allows determination of the initially unknown depth $L$:

$$0 = \frac{-2*c_{CH4} * exp\left(\sqrt{\frac{k_d}{D_{CH4}}}L\right) * \sqrt{\frac{k_d}{D_{CH4}}} * exp\left(-\sqrt{\frac{k_d}{D_{CH4}}}L\right)}{\left[-exp\left(-\sqrt{\frac{k_d}{D_{CH4}}}L\right) + exp\left(\sqrt{\frac{k_d}{D_{CH4}}}L\right)\right]} - F_{CH4} \tag{9}$$

Once $L$ is known total $CH_4$ uptake can be calculated from:

$$J_{CH4} = -D_{CH4} * \frac{dCH4}{d_z}|_{z->z=0} = -D_{CH4}\left(-A\sqrt{\frac{k_d}{D_{CH4}}} + B\sqrt{\frac{k_d}{D_{CH4}}}\right) \tag{10}$$





Where $A$ and $B$ are defined by Eqs. (6) and (7). When $L$ tends to infinity Eq. (10) is equivalent to C07 solution; however, Eq. (10) also allows for (i) complete consumption of $CH_4$ within the soil interval, and (ii) influx of $CH_4$ from beneath the soil profile (*e.g.*, from thawing permafrost or production of $CH_4$ in oxygen-depleted soils).

Figure 1 illustrates $CH_4$ soil profiles and the penetration depth of $CH_4$ into soil, $L$, for different $k_d$ values, $F_{CH4} = 0$

5 and $D_{CH4} = D_{0CH4}$ (diffusivity in free air) (Table1). It is expected that $L$ will vary spatially depending on local $k_d$, $D_{CH4}$ and soil properties.

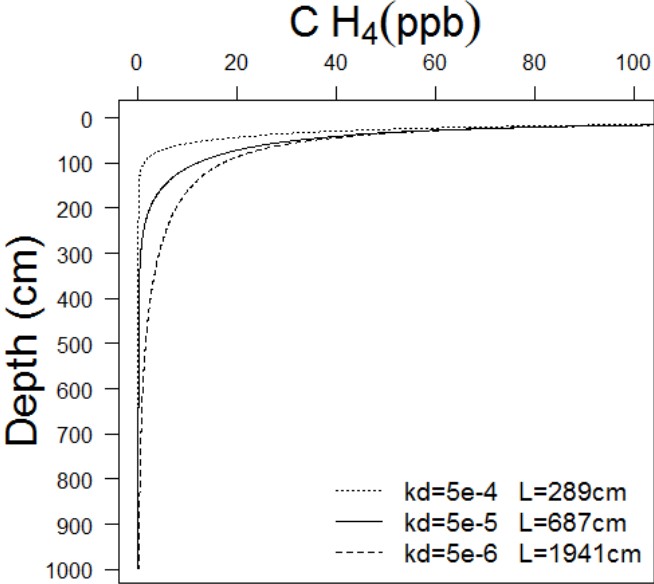

**Figure 1: Computational solution of Eq. (9) for different values of $k_d$. Parameter $L$ is defined as the depth where 99.9% of $CH_4$ in soil pore spaces is removed.**

**Table 2. Descriptions of four soil methanotrophy models.**

| Model / Study | Description | $CH_4$ uptake calculation ($J_{CH4}$) | Eq. |
|---|---|---|---|
| P96<br>Potter et al. (1996) | Model based on Fick's first law. The calculation of the uptake flux is approximated numerically and based on the diffusion of $CH_4$ into soil. | $J_{CH4} = D_{CH4} \frac{\Delta C_{CH4}}{\Delta_z}$ | (11) |
| R99<br>Ridgwell et al. (1999) | R99 extends the P96 model by including an explicit term for microbial oxidation of $CH_4$ in soil. The uptake flux is approximated numerically, using Fick's first law and adopting a first | $J_{CH4} = \frac{C_{CH4} D_{CH4}}{z_d} \left( 1 - \frac{D_{CH4}}{D_{CH4} + k_d z_d} \right)$ | (12) |



| | | |
|---|---|---|
| | order rate law for microbial oxidation, assuming that oxidation occurs in a thin $\epsilon$ cm layer located at 6 cm depth. | |
| C07 Curry (2007) | C07 adopts the diffusion-reaction equation that underlies R99. However, C07 solves the equation analytically (as opposed to semi-numerically). The model also improves representation of soil moisture influence on the microbial oxidation rate. C07 refines methanotrophy response at subzero temperatures on the basis of observations. | $$J_{CH4} = C_{CH4} r_N r_w \sqrt{D_{CH4} k_d} \qquad (13)$$ |
| MeMo This study | Incorporates a general mathematical description of $CH_4$ uptake flux, allowing for complete consumption of $CH_4$ at an initially unknown depth $L$ and $CH_4$ flux through the lower boundary. Refines representation of the influence of soil moisture, temperature and nitrogen deposition on $CH_4$ oxidation. | $$J_{CH4} = -D_{CH4}\left(-A\sqrt{\frac{k_d}{D_{CH4}}} + B\sqrt{\frac{k_d}{D_{CH4}}}\right) \quad (10)$$ |

MeMo is based on the more general solution (Eq. (10)) and uses local methanotrophy rates ($k_d$) and diffusion coefficients ($D_{CH4}$) based upon soil conditions to determine $CH_4$ penetration depths ($L$). We assume no *in situ* production of $CH_4$ or upward $CH_4$ flux from below (*i.e.*, $F_{CH4} = 0$) because of a scarcity of field data for model validation. However, a flux from below can be employed in MeMo to enable a more comprehensive quantification of soil $CH_4$ uptake that also potentially accounts for consumption of upward migrating $CH_4$ and autochthonous $CH_4$ produced in oxygen-depleted microsites of finely textured soil.

## 2.3 Parameters

The rate of $CH_4$ uptake by soil is controlled by the balance between gaseous diffusion of atmospheric $CH_4$ into soil and the rate of $CH_4$ oxidation by methanotrophic bacteria as described by Eq. (14) and Eq. (20), respectively. Thus, $D_{CH4}$ and $k_d$ are key parameters and accurate characterization of their values is essential for robust quantification of the soil $CH_4$ sink.





### 2.3.1 Soil CH$_4$ Diffusivity, $D_{CH4}$

Similar to the R99 and C07 model, $D_{CH4}$ in MeMo is determined from the diffusivity of CH$_4$ in free air ($D_{0CH4}$; Table 1) adjusted for the influence of temperature ($G_T$) and soil structure ($G_{soil}$):

$$D_{CH4} = D_{0CH4} * G_T * G_{soil} \tag{14}$$

The gaseous diffusion coefficient of CH$_4$ in soil increases linearly with temperature T (°C) (Potter et al., 1996) according to the relationship:

$$G_T = 1.0 + 0.0055\, T\ (°C) \tag{15}$$

The soil structure factor ($G_{soil}$) accounts for the effects of pore size, connectivity and tortuosity on gaseous diffusion and is determined according to the parameterization of Moldrup et al. (1996):

$$G_{soil} = \Phi^{4/3}\left(\frac{\Phi_{air}}{\Phi}\right)^{1.5+3/b} \tag{16}$$

Where $\Phi$ is total pore volume (cm$^3$ cm$^{-3}$), $\Phi_{air}$ is air-filled porosity (cm$^3$ cm$^{-3}$) and $b$ is a scalar that accounts for soil structure. Total pore volume is defined as a function of bulk density $\rho$ (g cm$^{-3}$) and average particle density $d$ (Table 1) (Brady et al., 1999):

$$\Phi = 1 - \left(\frac{\rho}{d}\right) \tag{17}$$

The scalar $b$ in Eq. (16) is calculated as a function of soil clay content ($f_{clay}$; %) as proposed by Saxton et al. (1986):

$$b = 15.9\, f_{clay} + 2.91 \tag{18}$$

Air-filled porosity ($\Phi_{air}$) is determined from the difference between total pore volume and soil water content $\theta$ (%):

$$\Phi air = \Phi - \theta \tag{19}$$



### 2.3.2 Rate Constant for CH₄ Oxidation, $k_d$

The CH$_4$ oxidation rate ($k_d$) is defined as the base oxidation rate constant ($k_0$) for an uncultivated moist soil at 0° C scaled by three factors to account for the influence of soil moisture ($r_{SM}$), soil temperature ($r_T$), and nitrogen content ($r_N$):

$$k_d = k_0 * r_{SM} * r_T * r_N \qquad (20)$$

The R99 and C07 models used a similar equation to estimate $k_d$ but without the $r_N$ parameter, opting instead to employ intensity of agricultural activity as a proxy to account for the inhibitory effects of N deposition on soil methanotrophy. Moreover, model C07 excluded $r_N$ from the $k_d$ formulation and used a N deposition term to modify total CH$_4$ uptake flux (Table 2, Eq. 13), which results in a larger N inhibition effect. The approach employed in MeMo is to use N deposition data directly to modify $k_d$.

### 2.3.3 Base Oxidation Rate Constant, $k_0$

The base oxidation rate constant ($k_0$) is a key parameter that exerts significant control on $k_d$ and thus the estimated CH$_4$ uptake flux. For example, a 10-fold change in $k_0$ (and thus $k_d$) leads to a 3-fold decrease in the depth $L$ at which CH$_4$ is fully depleted from soil pores (Fig. 1) and a ~3-fold increase in total uptake of CH$_4$ (Fig. 2).

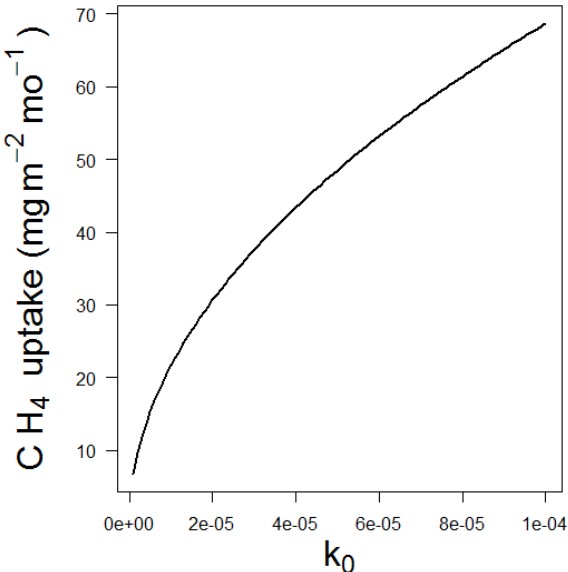

**Figure 2: Total CH₄ uptake for different values of $k_0$ (s$^{-1}$), assuming a constant value of $D_{CH4} = D_{0CH4}$ and no modification by soil temperature, moisture or nitrogen deposition.**





Rate constants can be defined either on the basis of theoretical considerations or through site-specific field and laboratory observations. Rates of soil microbial processes, such as $CH_4$ oxidation, are controlled by microbial biomass dynamics and community structure, and, thus a complex array of environmental factors, including temperature, substrate ($CH_4$) concentration, land use, moisture, pH and soil type (Ho et al., 2013). The influence of these environmental factors on

microbial $CH_4$ oxidation rates is not well characterized and thus all factors are not explicitly represented in models. Consequently, apparent rate constants implicitly account for some environmental factors via fitting field observations or laboratory experiments, resulting in parameter values that may be more environment- and model-specific. A possible limitation of such an approach is reduced transferability and predictive capacity in other environments or from a regional to the global scale. For example, Ridgwell et al. (1996) derived a single global estimate of $k_0 = 8.7 \times 10^{-4}$ s$^{-1}$ by fitting Eq. (12)

to 13 measured values of $J_{CH4}$, $D_{CH4}$ and soil temperature from four different studies. In contrast, Curry (2007) estimated a global $k_0$ of $5.0 \times 10^{-5}$ s$^{-1}$ based upon fitting Eq. (13) to a five-year time series of $J_{CH4}$ and soil temperature, moisture and $CH_4$ flux measurements from a single site in Colorado (Mosier et al., 1996). The order of magnitude difference in $k_0$ between the R99 and C07 models illustrates the potential model-specific nature of parameter values derived from experimental and observational data, as well as the limits and challenges for transferability. Soil methanotrophy is not unique in this regard

and parameterization of microbially mediated processes remains a common problem more generally in modelling approaches (*e.g.*, Arndt et al., 2013; Bradley et al., 2016).

Parameterization of $k_0$ in MeMo has been refined using recently time-series data published by Luo et al., (2013) and consisting of daily soil $CH_4$ uptake, temperature and soil moisture from three contrasting environments: temperate forest (Hoglwald Germany), tropical rainforest (Bellenden Ker Australia) and steppe (Inner Mongolia, China). The data sets were

used to explore potential variations in apparent $k_0$ values in different environments, including comparison with $k_0$ values from models R99 and C07. Data from each site were interpolated according to Eq. (10) to derive an apparent $k_0$ value for each biome. The $k_0$ values for temperate forest and steppe are similar to the $k_0$ value employed in the C07 model; however, the apparent $k_0$ for tropical forest is approximately three times smaller than the C07 $k_0$ value. The three newly derived $k_0$ values were employed in MeMo for their respective biomes and the $k_0$ value from the C07 model ($k_0 = 5.0 \times 10^{-5}$ s$^{-1}$) was

used for all other regions for which no biome specific $k_0$ values exist (Table 3). Similar $k_0$ values of $5.0 \times 10^{-5}$ s$^{-1}$ for temperate forest, steppe and short grass steppe indicate that this magnitude of $k_0$ is appropriate for many ecosystems. Yet, apart from the tropical wet forest, the data clearly indicate additional controls and the use of $k_0 = 1.6 \times 10^{-5}$ s$^{-1}$ will thus prevent an overestimation of simulated fluxes. Nevertheless, further research is required to better characterize this key parameter.

**Table 3. $k_0$ values from models R99 and C07 and new $k_0$ values employed in MeMo that were determined based upon temperate forest, tropical forest and steppe data from Luo et al. (2013).**

| Model | Biome | $k_0$ (s$^{-1}$) |
| --- | --- | --- |



| R99 | Global | $8.7 \times 10^{-4}$ |
|---|---|---|
| C07 | Global | $5.0 \times 10^{-5}$ |
| MeMo | Temperate forest | $4.0 \times 10^{-5}$ |
| | Tropical forest | $1.6 \times 10^{-5}$ |
| | Steppe | $3.6 \times 10^{-5}$ |
| | Other ecosystems | $5.0 \times 10^{-5}$ |

### 2.3.4. Soil Moisture Factor, $r_{SM}$

Both low and high soil moisture levels can negatively impact soil uptake of atmospheric $CH_4$ (Schnell and King, 1996; von Fischer et al., 2009). Scarcity of soil water generally inhibits soil microbial activity while excessive moisture attenuates gas diffusion, limiting entry of atmospheric $CH_4$ and $O_2$ into soil (Burke et al., 1999; McLain et al., 2002; McLain and Ahmann, 2007; West et al., 1999).

The R99 and C07 models incorporated parameters to address the limiting effects of low soil moisture levels on $CH_4$ uptake fluxes. The R99 model applied a soil moisture factor adopted from Potter et al. (1986) where $r_{SM}$ was calculated as a proportional ratio of precipitation ($P$) plus soil moisture ($SM$) divided by potential evapotranspiration ($ET$; Table 4, Eq. (21)). It was assumed that $r_{SM}$ decreases linearly when $(P+SM)/ET$ is less than one. The C07 model modified the response of soil methanotrophy to moisture using an empirical water stress parameterization and soil water potential based on findings from Clapp and Hornberger (1978) (Table 4, Eq. (22)). A consequence of that approach is that $r_{SM}$ decreases logarithmically to zero at an absolute soil water potential of $w < 0.2$ MPa (Fig. 3).

In MeMo, soil moisture (%) is used to calculate $r_{SM}$ and a formulation similar to the C07 model is used for low soil moisture values. A threshold of <20% soil moisture is applied because that value corresponds to optimum conditions for $CH_4$ oxidation in soil (Castro et al., 1995; Whalen and Reeburgh, 1996) and because inclusion of a water stress parameter better captures $CH_4$ uptake flux in dry ecosystems (Fig. 3; Curry, 2007).

Establishing parameters to quantify the impact of excess moisture on soil methanotrophy has proven more challenging. The C07 model relied upon soil pore space characteristics in factor $G_{soil}$ (Eq. (16)) to account for decreased gas diffusion and limitation of $k_d$ at high soil moisture content. However, attenuation of gas diffusion is only one impact of high soil water content and it is necessary also to account for the inhibitory effects of excessive moisture on $k_d$ (Boeckx and Van Cleemput, 1996; Dasselaar et al., 1998; Visvanathan et al., 1999). Soil moisture content >20% reduces $CH_4$ uptake due to a restricted diffusion of $CH_4$ and supply of $O_2$. The R99 and C07 models assume that microbial $CH_4$ oxidation remains active at a soil moisture content of 80%, an assumption that contradicts field investigations, which show that $CH_4$ uptake decreases rapidly at soil moisture levels >50% (Dasselaar et al., 1998). Thus, the soil moisture factor employed in MeMo also accounts for limitation of microbial $CH_4$ oxidation at a soil moisture content >20% after which rates of $CH_4$ uptake begin to decrease (Adamsen and King, 1993; Visvanathan et al., 1999). The $r_{SM}$ factor used in MeMo was determined by fitting a



Gaussian function to laboratory experimental data (Table 4, Eq. (23); Fig. 3a), following the approach of Grosso et al. (2000). The mean $r_{SM}$ and standard deviation determined using this approach were $0.2 \pm 0.2$.

**Table 4. Model R99, C07 and MeMo formulations for $r_{SM}$ response.**

| Model | Formulation | Eq. | Variable definitions |
|---|---|---|---|
| R99 | $r_{SM} = 1$ for P+SM/ETp>1 <br> $r_{SM} = P + SM/ETp$ for P+SM/ET p≤1 | (21) | P=precipitation <br> SM=soil moisture stored at 30 cm depth <br> ETp=potential evapotranspiration |
| C07 | $r_{SM} = 1$ for w<0.2MPa <br><br> $r_{SM} = \left[ 1 - \dfrac{log_{10}w - log_{10}(0.2)}{log_{10}(100) - log_{10}(0.2)} \right]^{0.8}$ for w≥0.2≤100MPa | (22) | w=saturation soil water potential |
| MeMo | $r_{SM} = \left[ 1 - \dfrac{log_{10}\frac{1}{SM} - log_{10}(0.2)}{log_{10}(100) - log_{10}(0.2)} \right]^{0.8}$ for SM<0.2 <br><br> $r_{SM} = \dfrac{1}{\sqrt[\sigma]{2\pi}} e^{-\frac{1}{2}\left(\frac{SM-0.2}{0.2}\right)^2}$ for SM>0.2 | (23) | SM=soil moisture |

A soil moisture factor ($r_{SM}$) was calculated for each set of observational data from independent field sites (Supplementary 1, Table S1) based upon an optimum rate of CH$_4$ uptake occurring at a soil moisture content of 20% ($r_{SM}$ = 1). The remaining $r_{SM}$ values were computed as a linear ratio of the CH$_4$ uptake rate at 20% water content. Figure 3b illustrates the pattern of response in methanotrophy rates to changes in soil moisture content in the R99, C07 models and

10  MeMo and the net effect on CH$_4$ uptake fluxes across a range of absolute soil moisture levels used to force parameter $r_{SM}$. The CH$_4$ uptake fluxes were calculated by varying soil moisture content while holding constant all other environmental parameters (temperature, $C_{CH4}$ and $N_{dep}$). The R99 and C07 models both predict greater CH$_4$ uptake fluxes than MeMo at soil moisture contents >20% with the R99 model yielding the highest flux rates; however, the C07 model and MeMo yield similar CH$_4$ uptake rates for much of the soil moisture range. Reduction of CH$_4$ uptake flux at high soil moisture levels due

15  to attenuation of gas diffusion cannot be managed solely through the term $G_{soil}$ (*i.e.*, reduction in free pore space). MeMo also accounts for inhibition of microbial CH$_4$ oxidation rates at elevated soil moisture content, predicting lower CH$_4$ uptake flux as a result of more realistic $r_{SM}$ values determined from the Gaussian response observed in field data from three different global biomes (Luo et al., 2013).



### 2.3.5. Temperature Factor, $r_T$

Temperature exerts an important influence on rates of microbial processes and consequently all models parameterize for the effects of temperature on soil methanotrophy. The R99 model employs a Q10 function derived from experimental data with a Q10 factor of 2 change over the temperature interval 0 to 15°C. The model assumes that bacterial methanotrophy ceases at temperatures <0°C (Table 5, Eq. (24)). The C07 model adopts the same Q10 factor as R99 for temperatures >0°C but employs a different response below 0°C. Soil water generally does not freeze at a surface temperature of 0°C and observations from cold regions provide ample evidence for the presence of methanotrophic activity at temperatures <0°C (Vecherskaya et al. 2013). The C07 model allows for a parabolic decrease of methanotrophy rates from 0 to -10°C (Table 5, Eq. (25)) based upon observations of $CH_4$ uptake in soil at subzero temperatures (Grosso et al., 2000).

Parameterization of a temperature factor ($r_T$) is revisited in MeMo based upon availability of new experimental data for soil from different biomes (Supplementary 1, Table S2). A Q10 factor having a value of 1.95 was determined for the temperature interval 0 to 15 °C by curve fitting and minimizing linear errors ($r^2 = 0.75$, p=1.9 $e^{-11}$; Table 5, Eq. (26)). The factor $r_T$ was determined by using the observed $CH_4$ uptake flux at 10°C at each site as the base of the Q10 function (Fig. 3c). An exponential decrease in $CH_4$ uptake flux was assigned to the temperature range 0 to -5°C as recommend by Castro et al. (1995) and Grosso et al. (2000). Moreover, the amount of frozen soil increases exponentially with decreasing temperatures (Low et al., 1968) and consequently, $CH_4$ uptake also should decline exponentially.

**Table 5. Model R99, C07 and MeMo formulations for $r_T$ response.**

| Model | T<0°C | T≥0°C | Eq. |
|---|---|---|---|
| R99 | $r_T = 0$ | $r_T = \exp(0.0693T - 8.56 \times 10^{-7} T^4)$ | (24) |
| C07 | $r_T = (0.1T + 1.0)^2$ if T>-10°C | $r_T = \exp(0.0693T - 8.56 \times 10^{-7} T^4)$ | (25) |
| MeMo | $r_T = 1/\exp(-T)$ | $r_T = \exp(0.1515 + 0.05238T - 5.946 \times 10^{-7} T^4)$ | (26) |

The pattern of change in the $r_T$ factor and $CH_4$ uptake flux for the temperature range -10 to 60°C is shown in Fig. 3d. The $CH_4$ uptake fluxes shown were calculated by varying temperature while holding other environmental factors constant (i.e., soil moisture, N deposition or agricultural land use, and $C_{CH4}$). All models exhibit an optimum in $CH_4$ uptake at 25°C characterized by a maximum $r_T$ and $CH_4$ oxidation rate. The key differences between models are the behavior of $r_T$ at temperatures below 0°C and the amplitude of response curves. The R99 model assumes that methanotrophy activity ceases at 0°C and consequently, $CH_4$ uptake rates decrease sharply at that temperature. In contrast, models C07 and MeMo both allow for methanotrophy at temperatures <0°C. In general, the exponential decrease of $r_T$ employed in MeMo more closely resembles natural patterns of soil methanotrophy at subzero temperatures than the parabolic decline employed in the C07 model consistent with observations reported by Castro et al. (1999) and Grosso et al. (2000). Although our



parameterization yields a fit similar to C07 to the limited observations available at temperatures <0°C the $r_T$ used in MeMo provides a simpler solution because it does not require multiple conditions to be met. In contrast, the C07 parameterization increases parabolically at temperatures <-10º C, which requires an additional condition to be incorporated into the model to prevent increased rates of CH$_4$ uptake at very low temperatures. Soil CH$_4$ uptake fluxes predicted by the C07 model are

greater than those calculated using MeMo because of the different parameterization at temperatures <0°C. Finally, the amplitude of the temperature response curve is greater and similar in models C07 and MeMo compared to model R99, in particular, at temperatures >25°C as a result of differences in the formulation and solution for CH$_4$ uptake flux (Fig. 3d).

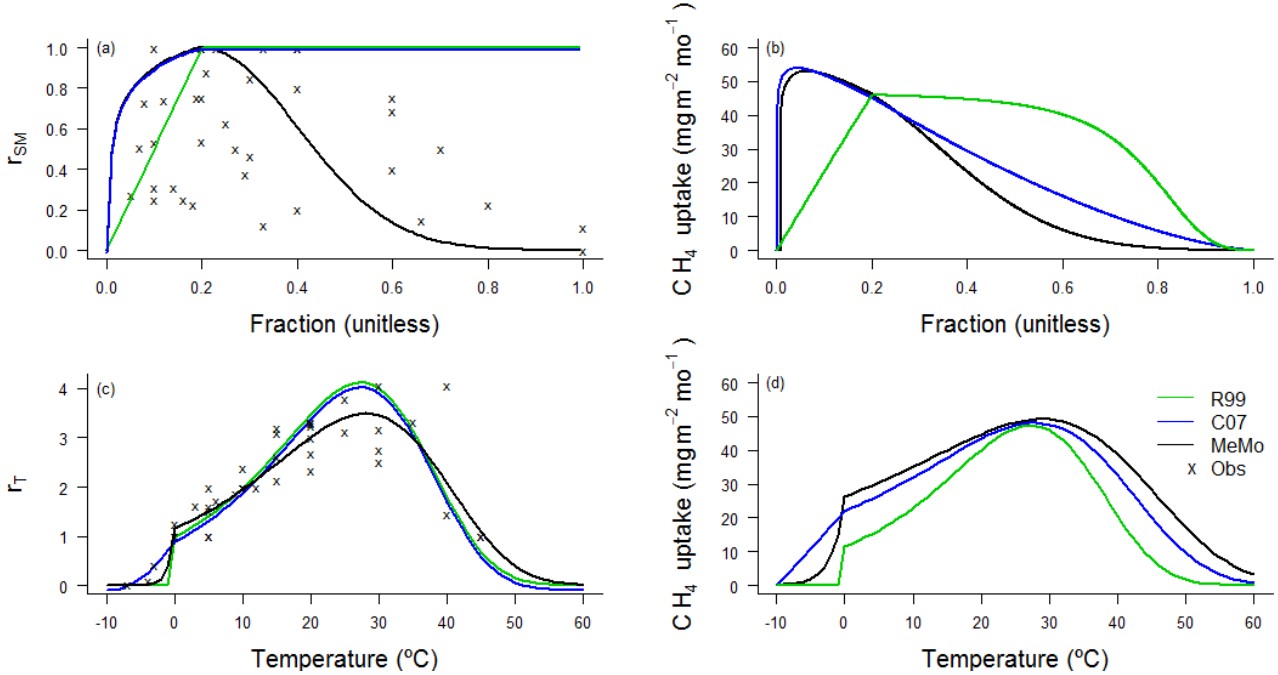

**Figure 3: CH$_4$ uptake response factors (a, c) and uptake fluxes (b, d) as a function of soil moisture ($r_{SM}$) and temperature ($r_T$).**
**Observations (shown as crosses) ($r_{SM}$, Supplementary 1, Table S1; $r_T$, Supplementary 1, Table S2), MeMo (black line), C07 (blue line) and R99 (green line).**

**2.3.6 Nitrogen Deposition factor, $r_N$**

The effect of nitrogen (N) deposition on CH$_4$ uptake is not as well constrained as the effects of temperature and soil
moisture. In general, field observations have shown that CH$_4$ consumption rates and thus, uptake fluxes decrease with N additions (Aronson and Helliker, 2010; Butterbach-Bahl and Papen, 2002; Steinkamp et al., 2001). Different processes have been suggested to explain this negative effect. Firstly, methanotrophs and ammonia oxidizers are capable of switching substrates (although the latter microorganisms typically consume N compounds preferentially if available) and therefore the presence of N compound reduces CH$_4$ consumption (Bradford et al., 2001; Gulledge and Schimel, 1998; Phillips et al., 2001;





Wang and Ineson, 2003; Whalen, 2000). In addition, intermediate and end products from methanotrophic ammonia oxidation (*i.e.*, hydroxilamida and nitrite) can be toxic to methanotrophic bacteria (Bronson and Mosier, 1994; MacDonald et al., 1996; Sitaula et al., 2000). Finally, large amounts of mineral fertilizers (*i.e.*, ammonium salts) can induce osmotic stress in methanotrophs inhibiting $CH_4$ consumption (Whalen, 2000). However, other studies suggest a positive effect of N

fertilization on $CH_4$ oxidation rates. One of the mechanism invoked to explain the positive effect is a stimulation of nitrifying bacteria to consume $CH_4$ by increased inputs of N due to an improvement in living conditions (Cai and Mosier, 2000; De Visscher and Cleemput, 2003; Rigler and Zechmeister-Boltenstern, 1999). The positive effect of N addition on $CH_4$ oxidation rates has been observed primarily under experimental conditions and also greatly depends on the local microbial community structure. Therefore, we assumed that N has an inhibitory effect on uptake of atmospheric $CH_4$ in all scenarios.

The C07 and R99 models both account for the negative effect of N inputs on $CH_4$ uptake fluxes via a N deposition factor $r_N$. In model R99, $r_N$ directly affects $k_d$ while in model C07 $r_N$ directly modifies the uptake flux. Both models parameterize the negative effect of N inputs on $CH_4$ oxidation rates as a function of agricultural intensity (as a fraction of area) as a proxy for fertilizer application (Table 6, Eq. (27)). However, the mathematical description of $r_N$ used by R99 and C07 does not account for the enhanced N deposition by anthropogenic activity because its global distribution was not well

known at the time of model development. Here, we suggest a mathematical description of $r_N$ that accounts for all anthropogenic N deposition sources: fossil fuel combustion, biomass burning and fertilizer application (Lamarque, 2013).

    The computation of $r_N$ in MeMo is a function of: i) the inhibitory effect on $CH_4$ uptake, and ii) the distribution and amount of N deposition in soil (Zhuang et al., 2013). We estimated the percent reduction of $CH_4$ uptake per mol of N added based on field and laboratory observations (Supplementary 1, Table S3). We determined an average inhibition $\alpha$ of 0.33

%mol $N^{-1}$ based on the mean uptake reduction per mol of N added. The N response function $r_N$ was governed by Eq. (29):

$$r_N = 1 - (N_{soil} * \alpha) \qquad\qquad (29)$$

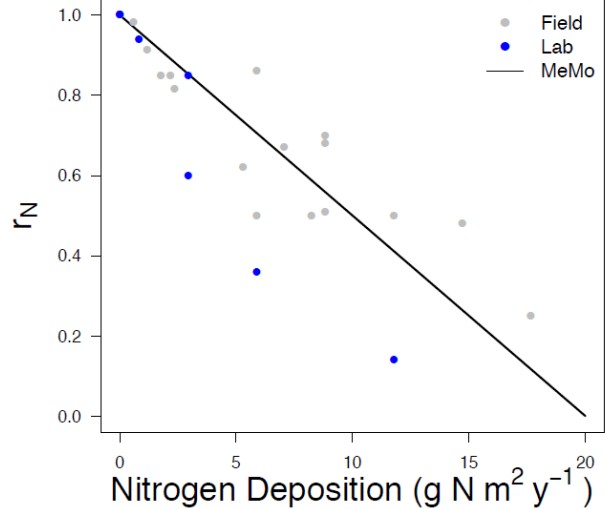



**Figure 4: CH$_4$ uptake response as a function of nitrogen deposition factor $r_N$. The linear fit (black line) is based on observations from field and laboratory measurements (gray and blue dots; Supplementary 1, Table S3).**

In cases where entry of N deposition into soil is limited by bulk density $\rho$, ninety percent of N compounds tend to remain at depths $z <= 5$ cm before exponentially decreasing in concentration with depth (Schnell and King, 1994). Thus, $N_{soil}$ was calculated as N deposition (kg N ha$^{-1}$ y$^{-1}$) divided by $\rho$ at $z = 5$cm:

$$N_{soil} = \frac{Ndep}{(\rho * z)} \tag{30}$$

Figure 4 shows the change in $r_N$ in relation to N deposition rate and the form of Equation (29).

**Table 6. Model R99, C07 and MeMo formulations for $r_N$ response.**

| Model | Formulation | Eq. | Driving data | |
|-------|-------------|-----|--------------|---|
| R99 | $r_N = 1.0 - (0.75 \times I)$ | (27) | I = fractional intensity of cultivation | |
| C07 | $r_N = 1.0 - (0.75 \times I)$ | (28) | I = fractional intensity of cultivation ($r_N$ outside of $k_d$ parameterization) | |
| MeMo | $r_N = 1 - (N_{soil}) * \alpha$ | (29) | $N_{soil} = \frac{Ndep}{(\rho * z)}$ | (30) |

### 3.0 Model implementation

MeMo was implemented in R (version 3.0.1) and simulations were carried out with a spatial resolution of 1°x1° and a monthly temporal resolution for the period between 1990 and 2009. The model code, a simple model case study for year 2000 and output for 1990-2009 are available as a supplement to this manuscript. To enable model-model comparisons and assess the combined effect of all refinements introduced in MeMo on the global CH$_4$ uptake flux estimate, the R99 and C07 models were also implemented in R at identical spatial and temporal resolutions and forced using the same driving data.

### 3.1 Forcing data

MeMo, the C07 and the R99 model were forced using global, monthly observations of soil moisture, temperature, atmospheric CH$_4$ concentrations, N deposition, soil bulk density, and clay content for the period 1990-2009. Information about data sources and maps of the forcing data are provided in Supplementary 2.





Satellite observations of soil moisture at a spatial resolution of 1x1° and a monthly temporal resolution are available for the period 1990-2009 from Dorigo et al. (2011); however, the data set contains gaps in some regions (*e.g.*, in areas of high-density vegetation). The use of MeMo as a predictive tool to estimate the past and future global $CH_4$ soil sink relies strongly on the use of soil moisture from standard climate models, such as output from land surface models or Dynamic

Global Vegetation Models (DGVMs). Therefore, gaps in the Dorigo et al. (2011) data set were filled using soil moisture data from an ensemble of 9 DGVMs (TRENDY; Sitch et al., 2015). The R99 model parameterizes the effect of soil moisture on $CH_4$ uptake fluxes as a function of precipitation and evaporation and therefore, R99 was forced using monthly data sets of precipitation (CRU3.1; Harris et al., 2014) and evapotranspiration (TRENDY; Sitch et al., 2015). Temperature forcing is constrained by global data sets for surface temperature as a proxy for soil temperature (CRU3.1; Harris et al., 2014).

Monthly mean global atmospheric $CH_4$ concentrations multiplied by the latitudinal atmospheric $CH_4$ gradient were calculated from Rigby et al. (2008). The N deposition data were obtained from an atmospheric chemical transport model embedded in an Earth System Model (Lamarque et al., 2013). Because the R99 and C07 models express the influence of N deposition on $CH_4$ uptake fluxes as a function of agricultural fraction rather than N deposition (see section 2.3.6), R99 and C07 were forced using annual global gridded land-use change data from Hurtt et al. (2011). Finally, global gridded

observations for bulk density and clay content were taken from Shangguan et al. (2014).

Areas that have less than 0.5% average annual soil moisture content were masked (*e.g.*, Sahara Desert) because it was assumed $CH_4$ uptake is negligible under such conditions. If the areas were left unmasked, then MeMo would overestimate $CH_4$ uptake across the regions due to high temporal variability in the driving data (*e.g.*, a month with no moisture followed by a month with >20%). Irregular short-lived precipitation events in deserts lead to unreliable estimates

of soil uptake of atmospheric $CH_4$ because such areas are unlikely to host well-established communities of methanotrophic bacteria capable of responding rapidly to short-term increases in soil moisture.

## 4.0 Results and Discussion

The following sections critically evaluate MeMo estimates of the global $CH_4$ sink (section 4.1), as well as the regional distribution of CH4 uptake and its main drivers (section 4.2) in the context of available field data and published model

predictions.

### 4.1 Global $CH_4$ Uptake by Soils

MeMo predicts an average annual global flux of $34.3 \pm 0.6$ Tg $CH_4$ y$^{-1}$ for the period 1990 to 2009. The estimated global uptake compares well with estimates from terrestrial ecosystem models, DGVMs and global atmospheric inversions (Table 7). Zhuang et al. (2013) determined a similar average global uptake flux of $34 \pm 2$ Tg $CH_4$ y$^{-1}$ during the 21$^{st}$ century using a

modified version of the Terrestrial Ecosystem Model (TEM) while Spahni et al. (2011) estimated an uptake flux of 38.9 Tg $CH_4$ y$^{-1}$ using the LPJ-WHyMe DGVM. Hein et al. (1997) predicted a similar flux through atmospheric inversions but with a greater level of uncertainty ($30 \pm 15$ Tg $CH_4$ y$^{-1}$). Upscaling of field measurements of soil methanotrophy rates from 120



different studies spanning a wide range of ecosystems yielded a value of $36 \pm 23$ Tg $CH_4$ y$^{-1}$ (Dutaur and Verchot, 2007). In contrast, flux estimates based upon extrapolation of long-term records of $CH_4$ uptake in a smaller number of soil types resulted in an estimated flux of 28.7 Tg $CH_4$ y$^{-1}$ (Dörr et al. 1993). Similarly, global extrapolation of measurements made solely on northern European soils yielded a sink strength of 29 Tg $CH_4$ y$^{-1}$ (Smith et al. 2000).

**Table 7. Global CH₄ uptake estimations**

| Methodology | Reference | Global uptake by soils (Tg $CH_4$ y$^{-1}$) |
|---|---|---|
| Observation | Dörr et al. (1993) | 28.7 |
| Observation | Smith et al. (2000) | 29 |
| Observation | Dutaur and Verchot (2007) | $36 \pm 23$ |
| Atmospheric inversions | Hein et al. (1997) | $30 \pm 15$ |
| Model (P96) | Potter et al. (1996) | $20 \pm 3$ |
| Model (R99) | Ridgwell et al. (1999) | $38.1 \pm 1.1$ |
| Model | Spahni et al. (2011) | 38.9 |
| Model (C07) | Curry (2007) | $29.3 \pm 0.6$ |
| Model | Zhuang et al. (2013) | $34 \pm 2$ |
| **Model (MeMo)** | **(This study)** | **$34.3 \pm 0.6$** |

The average annual soil sink for atmospheric $CH_4$ estimated by MeMo ($34.3 \pm 0.6$ Tg $CH_4$ y$^{-1}$) is greater than global uptake predicted using the P96 and C07 model ($20 \pm 3$ Tg $CH_4$ y$^{-1}$ and $29.3 \pm 0.6$ Tg $CH_4$ y$^{-1}$, respectively). The R99 model

predicts a global sink of $38.1 \pm 1.1$ Tg $CH_4$ y$^{-1}$, which compares more favorably with the MeMo estimate. The observed differences in mean global soil uptake of atmospheric $CH_4$ estimated using models R99, C07 and MeMo forced with identical data are attributed primarily to three factors: (i) their respective mathematical solutions of reaction-transport equations (section 2.2), (ii) differences in parameterization of $k_0$ (section 2.3.3), and (iii) differences in formulation of $r_N$ (section 2.3.6). The R99 model predicts soil uptake that is 10% and 24% greater, respectively, than fluxes estimated using

MeMo and the C07 model. These differences are due to the R99 model applying a $k_0$ that is one order of magnitude greater than $k_0$ values used in the C07 model and MeMo. The amplifying effect of the large $k_0$ is partially offset by the semi-numerical approximation (Eq. 12) employed in the R99 model, which results in the final global $CH_4$ uptake flux being of similar magnitude to the MeMo and the C07 model estimates. Finally, the low uptake predicted by the C07 model is a consequence of the parameterization of the nitrogen inhibition effect ($r_{N,}$) and its direct modification of the $CH_4$ flux rather

than the $CH_4$ oxidation activity ($k_d$) (section 2.3.3). Nitrogen inhibition was responsible for a global reduction in $CH_4$ uptake of 0.34 Tg y$^{-1}$ in MeMo compared to 7.3 and 2.3 Tg y$^{-1}$ in the C07 and R99 models, respectively.



## 4.2 Regional CH₄ Uptake by Soils

The latitudinal distribution of soil uptake rates of atmospheric CH$_4$ predicted using the R99 model, the C07 model and MeMo are shown in Fig. 5 accompanied by direct measurements of CH$_4$ oxidation rates from Dutaur and Verchot (2007) and a 10° running average. Figure 6 provides globally gridded maps of average simulated CH$_4$ uptake fluxes from the three models for the period 1990 to 2009 and a comparison between MeMo and the R99 and C07 models.

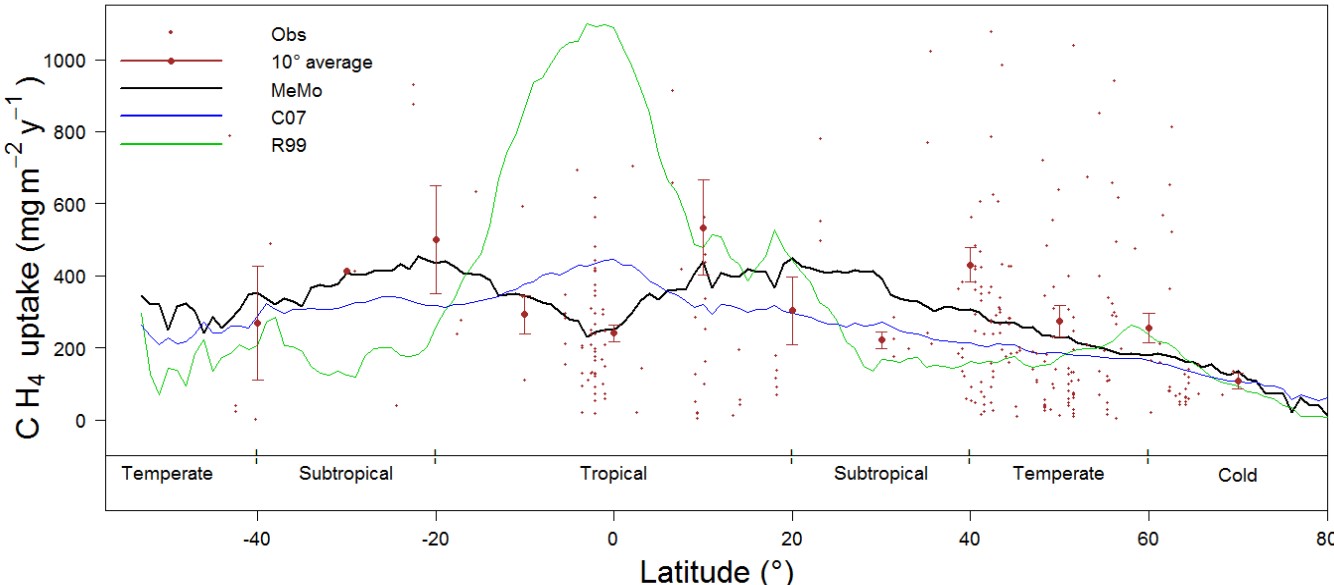

**Figure 5.** **Latitudinal distribution of the soil uptake predicted by models R99 (green line), C07 (blue line) and MeMo (black line). Measurements of CH$_4$ uptake (small brown dots; Dutaur and Verchot, 2007) and a 10º running mean of direct observations (large brown dots for average with bars representing one standard deviation error).**

The latitudinal distribution of observations reveals a scarcity of direct measurements of soil methanotrophy from sites in the southern hemisphere. Additionally, the frequency of measurements generally is low and rarely encompasses a full twelve-month period, which creates challenges for verifying model estimates of annual CH$_4$ uptake fluxes. Observations at specific latitudes typically exhibit a wide range of values, which are reflected in the large standard error bars calculated for the 10º running means (Fig. 5). Nevertheless, the averages of direct observations calculated for each 10° latitude interval show a distinct bimodal pattern with the lowest soil CH$_4$ uptake fluxes in the tropics and at high latitudes. Maximum rates of CH$_4$ uptake occur between 10 to 20° in both hemispheres (Fig. 5). MeMo simulates a similar bimodal latitudinal distribution of CH$_4$ uptake fluxes with an RSME that is 16.6 mg CH$_4$ m$^{-2}$ y$^{-1}$ lower than other models when fitted to 10° latitudinal averages of observational data. In contrast, the C07 and R99 models both predict a latitudinal distribution of soil methanotrophy that has CH$_4$ uptake maxima in equatorial regions and lower rates of CH$_4$ oxidation at mid-latitudes (~40°N and 20 to 40° S), resulting in higher RSMEs of 28 and 72 mg CH$_4$ m$^{-2}$ y$^{-1}$, respectively, when fitted to the 10° latitude



averaged data.   the R99 model significantly overestimates $CH_4$ uptake fluxes in the tropics (20°N to 20°S) and underestimates $CH_4$ oxidation in the subtropics (20 to 40° N and S), resulting in large differences for these regions relative to the MeMo simulations (Fig. 6e).  The C07 model predicts a latitudinal pattern of simulated $CH_4$ fluxes that is similar to R99; however, with much lower uptake fluxes in the tropics and no pronounced minima in the subtropics.  Consequently, the

RSME of the fit to observational data is much lower and regional differences relative to MeMo generally are smaller, ranging from 30% in the tropics to 20% in the subtropics (Fig. 6d).

The regional differences between MeMo and the R99 and C07 models result from differences in the parameterization of factors that govern $CH_4$ oxidation rates in the models: $k_0$, $r_{SM}$, $r_T$ and $r_N$.  The lower $k_0$ assigned to tropical wet forest (see section 2.3.3) accounts for the reduction in $CH_4$ uptake by tropical soil in MeMo.  The strong

agreement between MeMo simulation results and $CH_4$ uptake measurements presented in Fig. 5 suggests that the empirically derived lower $k_0$ value more accurately reflects soil $CH_4$ oxidation rates in the tropics.  However, we note the possibility that additional factors, or unexpected combinations of current factors, may influence rates of atmospheric $CH_4$ uptake in the tropics in ways that are not explicitly represented in the models.

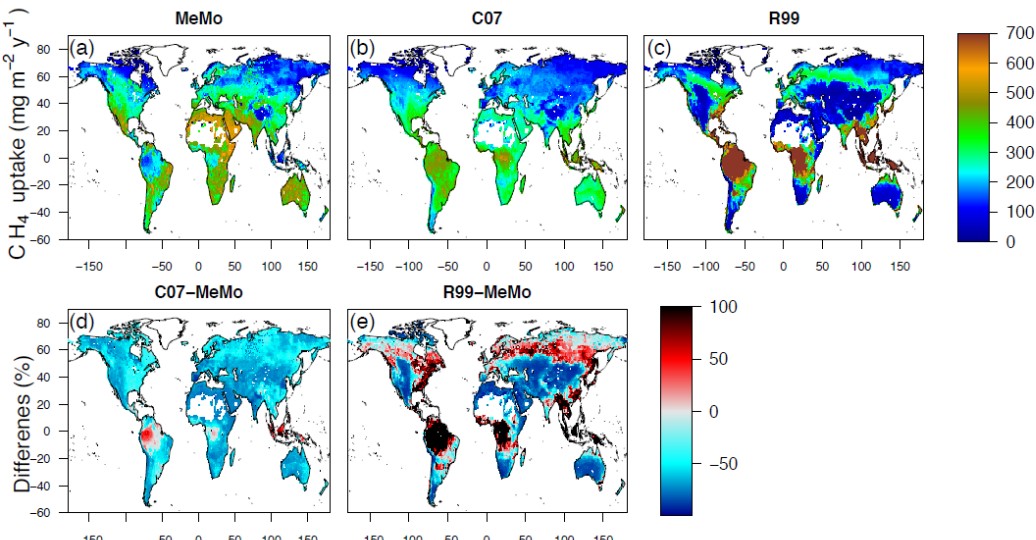

**Figure 6: Annual mean $CH_4$ uptake by soil predicted using models (a) MeMo, (b) C07 and (c) R99 for the period 1990-2009. Differences between models expressed in percent are shown in panels (d) C07 minus MeMo and (e) R99 minus MeMo.**

The influence of different environmental factors on soil $CH_4$ uptake was assessed by calculating the global $CH_4$ uptake flux while varying each factor (temperature, soil moisture and nitrogen deposition) independently and keeping other

factors constant (Figs. 7, 8 and 9).  Comparison of $r_{SM}$ values reveals large differences across models in tropical wet regions (Fig. 7), which explains the contrasting predictions of $CH_4$ uptake by MeMo (213 mg $CH_4$ $m^{-2}$ $y^{-1}$) versus models R99 (689 mg $CH_4$ $m^{-2}$ $y^{-1}$) and C07 (329 mg $CH_4$ $m^{-2}$ $y^{-1}$).  Formulation of $r_{SM}$ in MeMo (section 2.3.4) accounts for limitation of methanotrophic oxidation rates when soil moisture levels are >20% water content, a feature that is absent in the R99 and C07



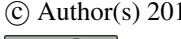

models.  In addition, the R99 model implements a linear decrease of $r_{SM}$ for soil moisture conditions <20%, which results in a 60 to 80% reduction in $CH_4$ oxidation rates in the subtopics.  The absence of this condition in models MeMo and C07 explains the significant differences in $CH_4$ uptake fluxes in subtropical regions (Figs. 5 and 6).

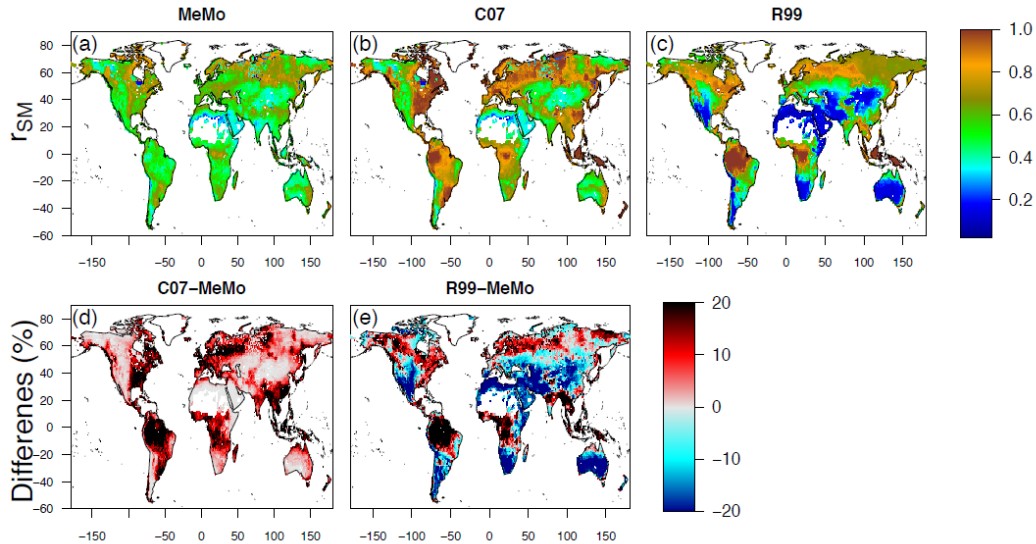

**Figure 7. Soil moisture response ($r_{SM}$) of $CH_4$ oxidation simulated by models (a) MeMo, (b) C07 and (c) R99.  Differences in model response expressed in percent are shown in panels (d) C07 minus MeMo, and (e) R99 minus MeMo.**

Formulations of $r_T$ are similar in the three models (section 2.3.5) and consequently, gridded maps of simulated $r_T$ values exhibit broadly similar global patterns in which high $r_T$ values are present at warm low latitudes and low $r_T$ values are predicted at cold high latitudes.  Notably, MeMo generally simulates $r_T$ values that are approximately 20% lower than those predicted by the C07 and R99 models (Fig. 8) because of the revised formulation of the $Q_{10}$ value.  MeMo and the C07 model simulate higher $r_T$ values than R99 at high latitudes because of differences in parameterization of $r_T$ at temperatures near 0°C.



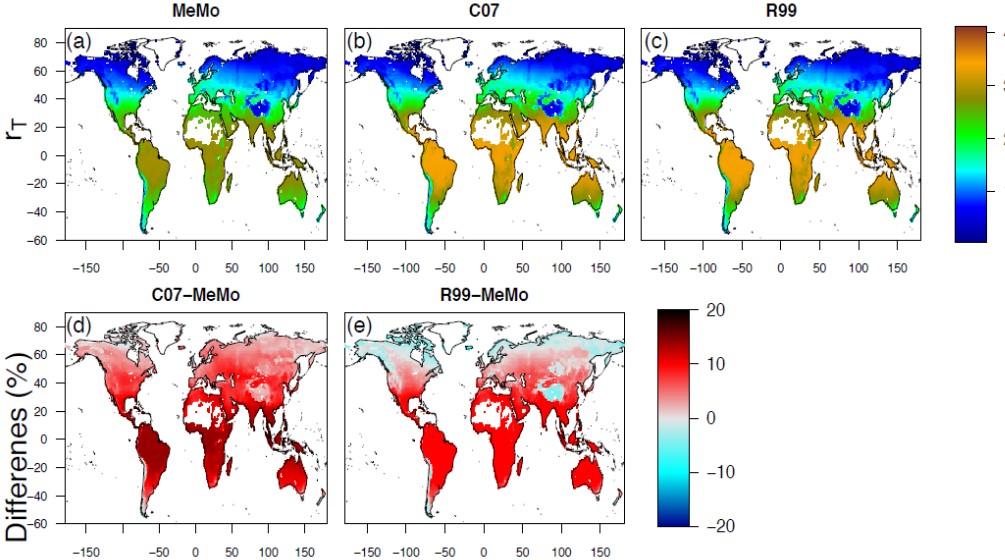

**Figure 8: Temperature response ($r_T$) of soil methanotrophy simulated by models (a) MeMo, (b) C07, and (c) R99. Differences in model response expressed in percent are shown in panels (d) C07 minus MeMo, and (e) R99 minus MeMo.**

Inhibition of soil methanotrophy due to N deposition ($r_N$) differs significantly between the three models. Nitrogen inhibition of $CH_4$ oxidation rates is lower in MeMo compared to the R99 and C07 models, in particular, at mid-latitudes (Fig. 9). The R99 and C07 models formulate $r_N$ as a function of agricultural intensity in contrast to MeMo which uses modelled N deposition. The difference in approach results in an ~20% higher $r_N$ factor in MeMo across most regions with the exception of high latitude areas (Fig. 9). In regions of intense agricultural activity and high N deposition (5 kg N ha$^{-1}$), such as Europe or the mid-western USA, the R99 and C07 models predict a reduction of up to 60% in $CH_4$ oxidation rates compared to a decrease of only ~10% simulated in MeMo. Two recent studies reported that high rates of N deposition (10 kg N ha$^{-1}$ y$^{-1}$) can reduce soil uptake of atmospheric $CH_4$ by ~8.6% (Fang et al., 2014; Zhang et al., 2008), which is consistent with the lower $r_N$ values employed in MeMo. The key limitation of the N effect approach adopted in the R99 and C07 models is the generalization of N inhibitory effects across different agricultural areas, crops and types of management, which results in excessive attenuation of $CH_4$ oxidation rates. For example, the R99 and C07 models predict a 75% reduction in $CH_4$ uptake rates for areas characterized by an agricultural intensity of 100%; however, Veldkamp et al. (2001) reported that inhibition of soil methanotrophy at such levels requires N addition >300 kg N ha$^{-1}$ y$^{-1}$, a loading rate which greatly exceeds N input levels typically associated with agriculture. Although a more conservative $r_N$ factor is employed in MeMo, the importance of accurate characterization of the attenuating effects of N addition on soil methanotrophy highlights the need for additional efforts to verify and refine parameterization of this key factor.





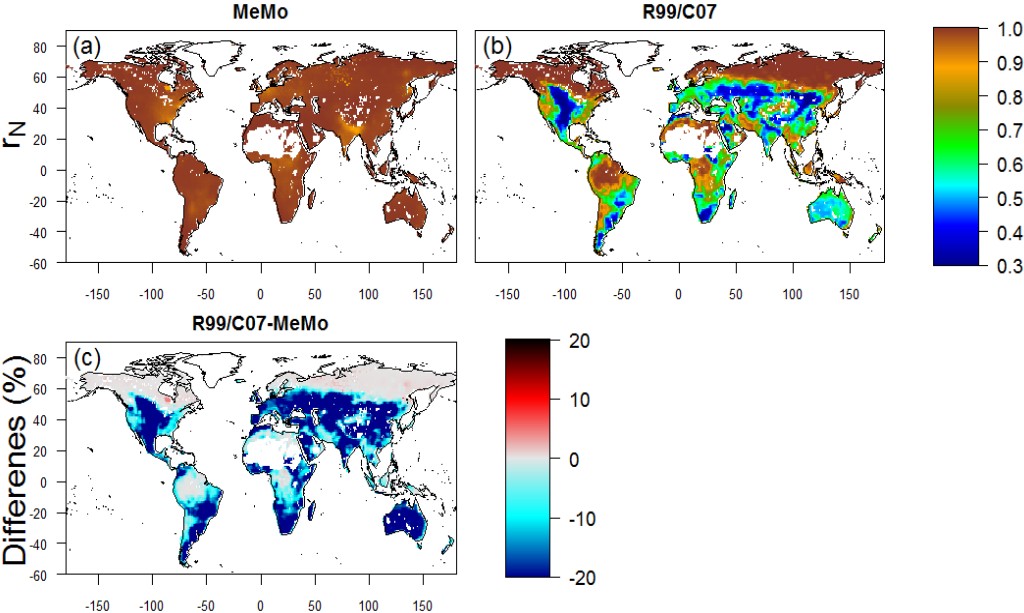

**Figure 9. Response of soil methanotrophy to nitrogen effect ($r_N$) simulated by models (a) MeMo, (b) R99, and C07. The responses for models R99 and C07 are both shown in panel (b) because they have the same formulation. The difference in model response between models R99/C07 minus MeMo expressed in percent is shown in panel (c).**

## 4.3 Temporal and Spatial Variability of Soil CH₄ Uptake

Field observations of soil uptake of atmospheric $CH_4$ are generally sparse both spatially and temporally. Consequently, our quantitative understanding of $CH_4$ uptake fluxes across different ecosystems and seasons is limited. Models provide a means to quantitatively explore spatial and temporal patterns of soil methanotrophy on scales that cannot be readily captured by field-based observations. Therefore, once tested and validated (see section 4.2), MeMo was used to quantitatively assess the variability of soil $CH_4$ uptake in different climate zones and ecosystems on seasonal time scales.

### 4.3.1 Regional Variability

The relative contribution of soil in each climatic zone to global uptake of atmospheric $CH_4$ as predicted by MeMo is summarized in Table 8. Soil in the northern hemisphere is estimated to account for approximately two thirds (67%) of the total global sink for atmospheric $CH_4$ because of the uneven distribution of landmasses between the northern and southern hemispheres. Notably, terrestrial areas in the northern subtropical and temperate zones collectively account for ~46% of the global soil sink for atmospheric $CH_4$. The southern tropical zone contributes a further ~19% to soil uptake of $CH_4$. The southern subtropical and northern tropical zones are estimated to contribute almost equally (~14%) to total $CH_4$ uptake (Table 8). The smallest proportion of soil $CH_4$ oxidation occurs in the southern temperate (0.5%) and northern polar (6%) zones due to a combination of small land area and low rates of $CH_4$ uptake. Model predictions of $CH_4$ uptake by climatic



zone provides insights into the relative importance of each region in the global $CH_4$ cycle but additionally begins to facilitate analysis of potential responses of the soil $CH_4$ sink within each zone to global change both due to climate and land management.

**Table 8. MeMo $CH_4$ uptake estimates by region.**

| Regions | Regional gridded mean (mg $CH_4$ $m^{-2}$ $y^{-1}$) | Total land area ($10^{12}$ $m^{-2}$) | Total $CH_4$ uptake (Tg $CH_4$ $y^{-1}$) | % of total |
|---|---|---|---|---|
| Cold zone (60°-90° N) | 102 | 18.7 | 2.2 | 6.4 |
| Temperate zone (40°-60° N) | 217 | 31.0 | 6.9 | 20.1 |
| Subtropic zone (20°-40° N) | 344 | 26.4 | 9.1 | 26.5 |
| Tropical zone (0°- 20° N) | 313 | 15.1 | 4.7 | 13.7 |
| **Northern Hemisphere Total:** | | **91.2** | **22.9** | **66.7** |
| | | | | |
| Temperate zone (40°-60° S) | 212 | 1.1 | 0.2 | 0.5 |
| Subtropic zone (20°-40° S) | 385 | 13.3 | 4.7 | 13.7 |
| Tropical zone (0°- 20° S) | 322 | 20.8 | 6.5 | 18.9 |
| **Southern Hemisphere Total:** | | **35.2** | **11.4** | **33.3** |

Further analysis of soil $CH_4$ uptake by ecosystem types (Table 9) shows that the highest gridded mean rates of $CH_4$ oxidation are associated with tropical deciduous forests (630 mg $CH_4$ $m^{-2}$ $y^{-1}$). The relatively low soil moisture content during the dry season (Supplementary 2, Figure S3) and the consistently high mean annual temperature (Supplementary 2, Figure S6) in such ecosystems promote high rates of soil methanotrophy. Furthermore, the soil typically possesses a low
clay content (Supplementary 2, Figure S2), which results in higher porosity that enhances gas diffusion and promotes higher rates of $CH_4$ oxidation. In comparison, rates of $CH_4$ uptake by soil in open and dense shrubland, temperate evergreen forest and savanna ecosystems (Table 9) are ~100 mg $CH_4$ $m^{-2}$ $y^{-1}$ lower but still highly significant globally.

Dense and open shrubland are characterized by constant climatic conditions (temperate and relatively low soil moisture; Supplementary 2 Figures S6 and S3, respectively) throughout the year, which in combination with a texture that
typically is sandy results in high annual $CH_4$ uptake rates (Tate et al., 2007). In contrast, high annual rates of $CH_4$ uptake in temperate evergreen forests result from elevated rates of soil methanotrophy during summer months (section 2.3.4), indicating that temperature is a key driver of $CH_4$ oxidation in such ecosystems (Borken et al., 2006; Ueyama et al., 2015; Wang and Ineson, 2003). Savannas share many climatic conditions with tropical deciduous forests but also commonly experience wildfire during the dry season. Both ecosystem types though are characterized by a marked seasonality driven by
the presence or absence of precipitation in combination with a consistent high mean annual temperature (Supplementary 2, Figure S6 and S3), which collectively support high rates of $CH_4$ uptake by soil.

Tundra, taiga, polar desert and other ecosystem types that are common at high latitudes (Supplementary 2, Figure S9) are characterized by the lowest mean annual rates of soil methanotrophy (<108 mg $CH_4$ $m^{-2}$ $y^{-1}$) because of low temperatures throughout most of the year. MeMo also predicts low rates of $CH_4$ uptake in tropical humid forest (330 mg





CH$_4$ m$^{-2}$ y$^{-1}$) due to low rates of bacterial CH$_4$ oxidation and the negative impact of high soil moisture levels on gas diffusion (see section 2.3.5). The CH$_4$ uptake rates estimated by MeMo are consistent with field observations by Dasselar et al. (1998) and Luo et al. (2013) which indicate that excess soil moisture strongly attenuates CH$_4$ uptake rates across a range of ecosystem types.

Finally, the global significance of each ecosystem type as a CH$_4$ sink depends strongly on spatial extent as well as CH$_4$ oxidation rates. Open shrubland (18.7%), grassland and steppe (12.7%), and savanna (11.3%) are the most important ecosystem types contributing to the global CH$_4$ soil sink (~46% collectively; Table 9) in MeMo because of high mean rates of CH$_4$ uptake (339 to 571 mg CH$_4$ m$^{-2}$ y$^{-1}$) in combination with a large areal extent globally (14 x 10$^{12}$ to 23 x 10$^{12}$ m$^2$). This finding is similar to the estimation by Potter et al. (1996) that warm and relatively dry ecosystems, such as semi-arid

steppe, tropical savanna, tropical seasonal forest, and chaparral, account for 40% of soil uptake of atmospheric CH$_4$ globally. Moreover, Luo et al. (2013) reported the highest annual CH$_4$ uptakes rates in dry savanna as part of a long-term field investigation of soil methanotrophy in several ecosystem types. Singh et al. (1997) also observed CH$_4$ uptake rates that were higher in savannah than temperate forest. Although both model simulations and available field observations suggest these ecosystems are important global sinks for atmospheric CH$_4$ there is presently a dearth of field measurements for warm and

dry environments relative to temperate ecosystems.

**Table 9. MeMo CH$_4$ uptake estimates by ecosystem type from Ramankutty and Foley (1999) land cover classification.**

| Ecosystem type | Global gridded mean (mg CH$_4$ m$^{-2}$ y$^{-1}$) | Total land area (x10$^{12}$ m$^{-2}$) | Total CH$_4$ uptake (Tg CH$_4$ y$^{-1}$) | % of total |
|---|---|---|---|---|
| Tropical Deciduous Forest | 630 ± 59 | 4.2 | 1.7 | 5.0 |
| Dense Shrubland | 580 ± 104 | 6.1 | 2.4 | 6.9 |
| Open Shrubland | 571 ± 132 | 23.3 | 7.3 | 21.2 |
| Temperate Broadleaf Evergreen Forest | 526 ± 70 | 2.0 | 0.7 | 2.0 |
| Savanna | 504 ± 128 | 14.1 | 4.5 | 13.1 |
| Temperate Needleleaf Evergreen Forest | 359 ± 90 | 3.9 | 1.2 | 3.4 |
| Grassland/Steppe | 339 ± 92 | 15.8 | 4.3 | 12.5 |
| Temperate Deciduous Forest | 331 ± 68 | 5.2 | 1.4 | 4.0 |
| Tropical Evergreen Forest | 330 ± 44 | 12.5 | 2.5 | 7.2 |
| Boreal Deciduous Forest | 288 ± 119 | 5.7 | 1.5 | 4.3 |
| Boreal Evergreen Forest | 275 ± 96 | 9.1 | 2.4 | 7.0 |
| Mixed Forest | 187 ± 86 | 13.4 | 2.7 | 7.8 |
| Tundra | 179 ± 144 | 6.2 | 1.7 | 4.9 |
| Polar Desert/Rock/Ice | 108 ± 48 | 0.4 | 0.01 | 0.0 |
| **Total** | | **124.1** | **34.3** | **100** |



### 4.3.2 Seasonal Variability

Global annual uptake of atmospheric $CH_4$ by soil exhibits a marked seasonality that reflects the dominance of the northern hemisphere in soil methanotrophy. The highest simulated $CH_4$ uptake fluxes occur during June, July, August (JJA) (10.4 Tg $CH_4$) followed by September, October and November (SON) (10.2 Tg $CH_4$), March, April and May (MAM) (7.2 Tg $CH_4$),

and finally, December, January and February (DJF) (6.5 Tg $CH_4$) (Fig. 10).

Methane uptake in the cold and temperate regions of the northern hemisphere generally is characterized by the largest seasonality exhibiting an amplitude of 30 mg $CH_4$ m$^{-2}$ mo$^{-1}$. In these regions, modeled uptake of $CH_4$ by soil is controlled strongly by temperature and consequently, ecosystems common at these latitudes (*e.g.,* boreal, needle leaf, temperate deciduous, mixed forest, polar deserts/rock/ice and tundra) show pronounced seasonal trends (Fig. 11), which also

are evident in field measurements (*e.g*., Priemé and Christensen, 1997) and emphasized in local mechanistic models (*e.g*., Oh et al., 2016). These finding suggest that the soil $CH_4$ sink in such ecosystems may be more sensitive to future change as a result of global warming.

In contrast, soil methanotrophy in temperate regions in the southern hemisphere are characterised by a weaker seasonality having an amplitude of 17 mg $CH_4$ m$^{-2}$ mo$^{-1}$ due to the prevalence of grassland and steppe, which contrasts with a

dominance of forest in the northern hemisphere. Seasonality of soil $CH_4$ uptake fluxes is even more muted in tropical and subtropical environments (<10 mg $CH_4$ m$^{-2}$ mo$^{-1}$) because of favourable and stable environmental conditions. Tropical deciduous forest and tropical evergreen forest, which are common in these climate zones are characterized by relatively constant $CH_4$ uptake fluxes throughout the year (Fig. 11); however, MeMo predicts greater seasonality (20 mg $CH_4$ m$^{-2}$ mo$^{-1}$) of $CH_4$ uptake by soil in drier subtropical ecosystems, such as open shrubland, savanna and grasslands (Fig. 11) because of

seasonality in soil moisture.

Notably, northern temperate forest in summer (JJA) was the ecosystem and time period possessing the highest average monthly $CH_4$ uptake fluxes (137 mg $CH_4$ m$^{-2}$ mo$^{-1}$) simulated by MeMo. During the rest of the year, the largest soil sink for atmospheric $CH_4$ occurred in the southern hemisphere in tropical deciduous forest of central Africa (DJF, 132 mg $CH_4$ m$^{-2}$ mo$^{-1}$; MAM, 131 mg$CH_4$ m$^{-2}$ mo$^{-1}$; SON, 129 mg $CH_4$ m$^{-2}$ mo$^{-1}$). This finding is significant because field

observations of soil methanotrophy in northern temperate forest during summer are the measurements most commonly extrapolated to an annual basis, which may lead to a possible overestimation of global $CH_4$ uptake fluxes.





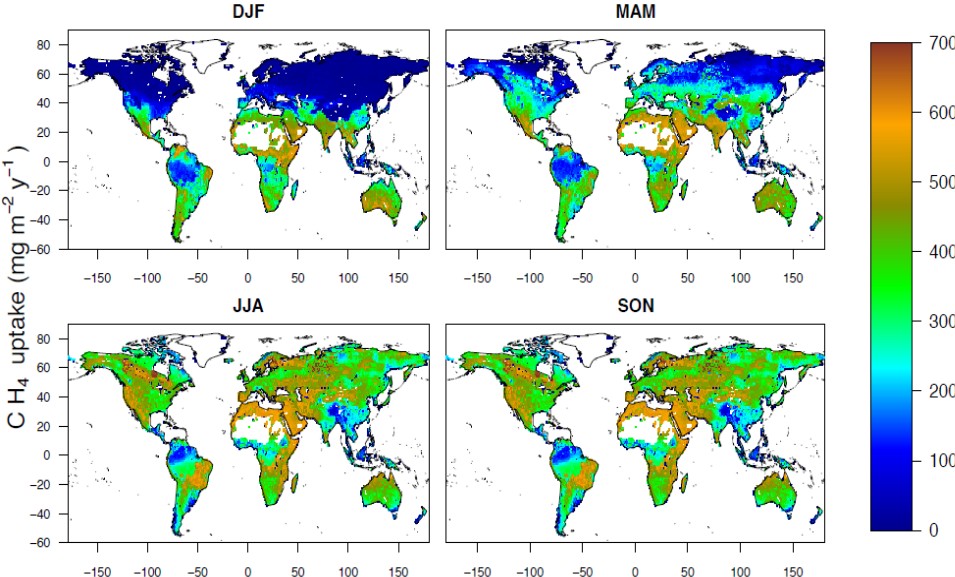

**Figure 10: Seasonal uptake of atmospheric CH₄ by global soils predicted by MeMo for the period 1990 to 2009.**





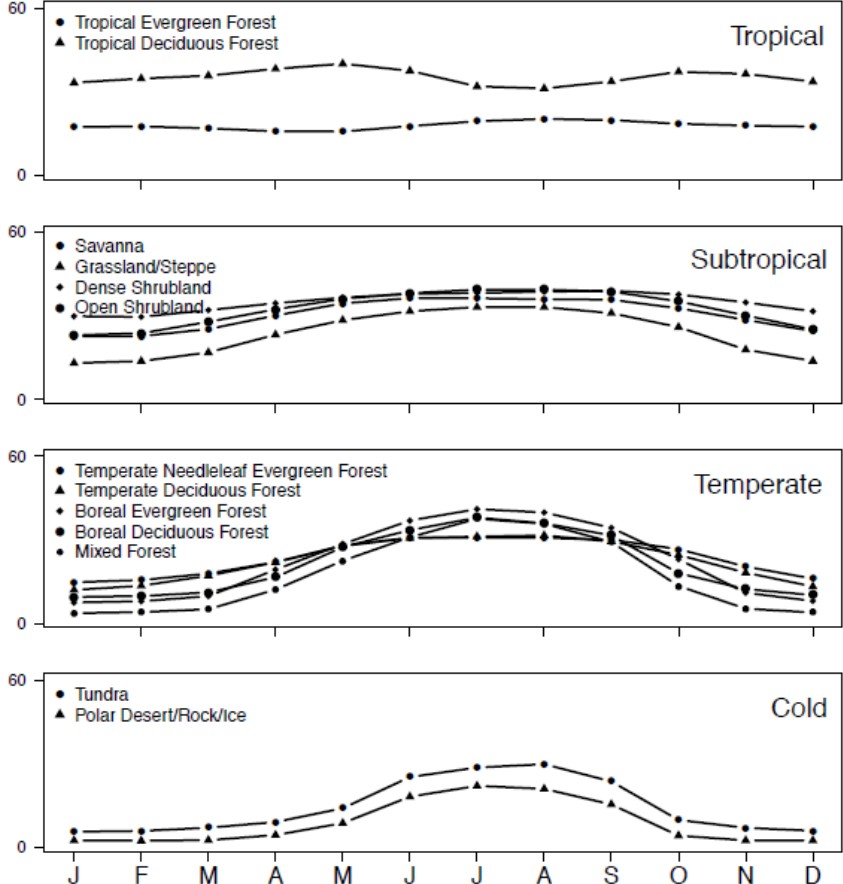

**Figure 11: Seasonal patterns of soil uptake of atmospheric CH4 by ecosystem for the four regions: cold, temperate, tropical and subtropical using MeMo model for the period 1990-2009.**

## 5 4.4 Model Limitations and Scope of Applicability

Several aspects of MeMo can be developed further, pending availability of additional field data, to improve estimation of global soil uptake of atmospheric $CH_4$. Firstly, the base oxidation rate of bacterial methanotrophy at 0°C ($k_0$) is a critical parameter necessary for accurate estimation of $CH_4$ uptake rates. There is presently a general dearth of published $k_0$ values for soil methanotrophy and moreover, ecosystem coverage is incomplete. Additionally, our parametrization for $k_d$ accounts

10 for methanotrophic activity in a one-dimensional soil matrix; however, other studies have separated $CH_4$ uptake in soil from methanotrophy in the rhizosphere to improve estimates of total $CH_4$ uptake (*e.g.*, Savrekov et al., 2016). This refinement has been modeled for local conditions but insufficient data about rhizosphere $CH_4$ oxidation rates prevent inclusion in MeMo and extension to a global scale.





Secondly, the $Q_{10}$ response of soil methanotrophy has been determined to date in only a small subset of ecosystems in which soils function as a sink for atmospheric $CH_4$. The majority of $Q_{10}$ values have been determined for bacterial oxidation of $CH_4$ under laboratory conditions and there is considerable variability in values across different ecosystems.

Thirdly, additional field observations of $CH_4$ uptake by soil are needed, in particular, long-term measurements at
individual sites that capture seasonality and inter-annual variability and from regions that presently have minimal or no representation (*i.e.*, the southern hemisphere, semi-arid ecosystems, etc.) in the current pool of observations.

Fourthly, additional observations and characterization of the effects of N deposition on soil methanotrophy are needed. The measurements ideally should be conducted *in situ* using N input rates that are appropriate for different environments and land use practices.
Finally, MeMo is parameterized to accommodate input of $CH_4$ from below (*i.e.*, subsurface methanogenesis or upward migration of deeply sourced $CH_4$); however, rigorous validation of that aspect of the model will require additional field observations, including better characterization of conditions under which $CH_4$ is produced in finely textured soils and deep sub-horizons. The presence, or periodic input, of high concentrations of $CH_4$ (*e.g.*, from permafrost melting) may impact competition for oxygen and niche space between low affinity $CH_4$-oxidizing bacteria and the high affinity
methanotrophs responsible for uptake of atmospheric $CH_4$. Refinement and validation of the capacity for MeMo to account for upward migrating or autochthonous $CH_4$ will enable the model to be used to estimate $CH_4$ flux from intermittently wet environments, which may currently fall outside the scope of process-based wetland models.

The process-based nature of MeMo and the breadth of conditions for which it has been validated provide scope for using the model to quantify $CH_4$ uptake in soil in a broad range of scenarios. For example, MeMo could be used to
determine global uptake of $CH_4$ by soil in the past during glacial or former interglacial periods. It may also be used to assess potential uptake rates of atmospheric $CH_4$ in future climate scenarios and further elevated tropospheric $CH_4$ mixing ratios.

**5.0 Conclusions**

We developed a processed-based model to simulate uptake of atmospheric $CH_4$ by soil, which was refined using newly reported experimental data and the introduction of recent insights into physical and biological mechanisms that drive soil
methanotrophy. We modified the general analytical solution proposed by Ridgwell et al. (1999) and Curry (2007) to account for a maximum depth of $CH_4$ uptake and to quantify upward migration and consumption of $CH_4$ produced *in situ*. Representation of the effects of nitrogen deposition, soil moisture and temperature on methanotrophy were improved based upon newly available data and recent advances in characterization of these processes. Finally, we proposed utilization of a different base oxidation rate $k_0$ for methanotrophy in different regions because its value changes in relation to environmental
conditions.

MeMo simulations produced a closer fit to observational data than two previous soil methanotrophy models (Ridgwell et al., 1999; Curry 2007). MeMo and observational data show a similar bi-modal latitudinal distribution of atmospheric $CH_4$ uptake by soil with the lowest fluxes at the equator and high-latitudes, and largest uptake fluxes at mid-





latitudes. Previous models simulated a dissimilar pattern with large uptake fluxes in equatorial regions, a difference that results primarily from improved representation of the soil moisture effect in MeMo.

MeMo simulations supported by observational data indicate that warm and semiarid regions are the most efficient soil sink for atmospheric $CH_4$. In these regions, tropical deciduous forest and dense open shrubland are characterized by

relatively low soil moisture and constant temperature during the year, which are key factors that promote high rates of $CH_4$ uptake by soil. In contrast, cold regions possessed the lowest $CH_4$ uptake rates, in particular, tundra and boreal forest, which have a marked seasonality driven by temperature, making soil methanotrophy in such areas potentially sensitive to future global climate change. The warm and wet tropical evergreen forest biome has $CH_4$ uptake rates that are ~50% less than warm and semiarid regions because excess soil moisture impacts soil-atmosphere gas exchange, resulting in a smaller $k_0$ (1.6

x10$^{-5}$ s$^{-1}$). The extensive area of shrubland, grassland, steppe and savanna globally yields a high total uptake of $CH_4$; however, there is presently a dearth of experimental data for these biomes and additional field observations are required to strengthen validation of MeMo simulations for these globally extensive areas.

MeMo simulations indicate that soil uptake of atmospheric $CH_4$ is reduced by as much as 10% in regions that receive high rates of nitrogen deposition. Globally, nitrogen deposition attenuates the soil sink for atmospheric $CH_4$ by 0.34

Tg yr$^{-1}$, which is an order of magnitude lower than previously reported because of the refined representation of the nitrogen inhibition effect in MeMo.

The accuracy of quantifying the modern soil sink for atmospheric $CH_4$ is improved using MeMo. In addition, the model can be used to explore changes in the relative importance of soil methanotrophy in the global $CH_4$ cycle in the past and the capacity of the soil sink to consume atmospheric $CH_4$ under future global change scenarios.

**Code and Data Availability**

MeMo was implemented in R (version 3.0.1). The model code and model output for 1990-2009 are available as a supplement to this manuscript. In addition, we also provide a post-processed driving dataset to run a simple, example model

case study for year 2000. All the forcing data used in this study are available from the following sources:

- temperature from CRU3.1, Harris et al. (2014): https://crudata.uea.ac.uk/cru/data/hrg/;
- vegetation mask from Ramankutty and Foley (1999): https://nelson.wisc.edu/sage/data-and-models/global-potential-vegetation/index.php;
- soil moisture from Dorigo et al. (2011) (Satellite): http://www.esa-soilmoisture-cci.org;
- soil moisture from TRENDY: (Sitch et al., 2015): http://www-lscedods.cea.fr/invsat/RECCAP/;
- nitrogen deposition from Lamarque et al. (2013): 1) https://www.isimip.org/gettingstarted/downloading-input-data/, 2) https://www.isimip.org/gettingstarted/details/24/; and
- clay content and bulk density from Shangguan et al. (2014): http://globalchange.bnu.edu.cn



*Author contributions.* FMF and SA developed and modified the model. FMF and GMT created the code. FMF ran the simulations, analysed the data and created all figures. All authors contributed to the interpretation of the results and
5   preparation of the manuscript.

*Competing interests.* The authors declare that they have no conflict of interest.

*Acknowledgements.* CONACyT Mexico is thanked for providing Ph.D. funding support to F. Murguia-Flores. S. Arndt acknowledges funding from the European Union's Horizon 2020 research and innovation programme under the Marie Sklodowska-Curie grant agreement No 643052 745 (C-CASCADES project). A.L. Ganesan is funded by a UK Natural Environment Research Council Independent Research Fellowship NE/L010992/1. G Murray-Tortarolo thanks the
Universidad Nacional Autonoma de Mexico and the University of Exeter for providing funding during his postdoctoral studies. We thank Guangjuan Luo and Klaus Butterbach-Bahl for providing methane uptake data from their long-term observational sites. The TRENDY modelling compendium are gratefully acknowledged for providing soil moisture data.

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
