# Peer review of "Soil Methanotrophy Model (MeMo v1.0): a process-based model to quantify global uptake of atmospheric methane by soil"

_Geoscientific Model Development, 2017_

## Referee Comment (RC1) · Anonymous Referee #1 · 3 Jul 2017

Review of "Soil Methanotrophy Model (MeMo v1.0): a process-based model to quantify global uptake of atmospheric methane by soil"

Summary
The manuscript describes a new model of methane consumption by upland soils. The topic of the modelling is hot and deserved a lot of attention in literature over recent years. Paper consists of detailed model description, data sources description, global and regional methane consumption estimates, their seasonal variability and discussion of the listed results.

Major comments
1. My first big concern is about correspondence between your solution of eq. (2) (using eqs. (3) and (6-7)) and boundary conditions. I will use the same symbols and introduce following substitution

$$\alpha = \sqrt{\frac{k_d}{D_{CH_4}}}$$

Upper boundary condition for your model is $CH_4(0) = C_{CH_4}$, Dirichlet type, fixed atmospheric level concentration. If we substitute eqs. (6) and (7) into eq. (3) and write it for the upper bound ($z = 0$) we get following equation:

$$CH_4(0) = \frac{C_{CH_4} \cdot \exp(\alpha \cdot L) \cdot \exp(-\alpha \cdot 0)}{(-\exp(-\alpha \cdot L) + \exp(\alpha \cdot L))} + \frac{C_{CH_4} \cdot \exp(-\alpha \cdot L) \cdot \exp(\alpha \cdot 0)}{(-\exp(-\alpha \cdot L) + \exp(\alpha \cdot L))} \qquad (r1)$$

Let's rewrite (r1) in the following form

$$CH_4(0) = C_{CH_4} \cdot \frac{\exp(\alpha \cdot L) + \exp(-\alpha \cdot L)}{(-\exp(-\alpha \cdot L) + \exp(\alpha \cdot L))} \qquad (r2)$$

To correspond your upper boundary condition fraction term $\frac{\exp(\alpha \cdot L) + \exp(-\alpha \cdot L)}{(-\exp(-\alpha \cdot L) + \exp(\alpha \cdot L))}$ from eq. (r2) should be equal to 1 (numerator and denominator should be the same and non-zero), but as we see, it is not.

Lower boundary condition for your model is $CH_4(L) = 0$, also Dirichlet type. By the way in several places you speak not about complete consumption of methane but 99.9% of $C_{CH_4}$ consumed. It creates ambiguity and should be fixed. If we substitute eqs. (6) and (7) into eq. (3) and write it for the lower bound ($z = L$) we get following equation

$$CH_4(L) = \frac{C_{CH_4} \cdot \exp(\alpha \cdot L) \cdot \exp(-\alpha \cdot L)}{(-\exp(-\alpha \cdot L) + \exp(\alpha \cdot L))} + \frac{C_{CH_4} \cdot \exp(-\alpha \cdot L) \cdot \exp(\alpha \cdot L)}{(-\exp(-\alpha \cdot L) + \exp(\alpha \cdot L))} \qquad (r3)$$

Let's rewrite (r3) in the following form

$$CH_4(L) = C_{CH_4} \cdot \frac{2}{(-\exp(-\alpha \cdot L) + \exp(\alpha \cdot L))} \qquad (r4)$$

It is obvious from this equation that $CH_4(L) \neq 0$. For example given in your paper parameter values $k_d = 5 \cdot 10^{-5}\ s^{-1}$; $D_{CH_4} = 0.196\ cm^2 \cdot s^{-1}$; $L = 687\ cm$ eq. (r4) leads to $CH_4(L) = 5.79 \cdot C_{CH_4}$. Not $0.001 \cdot C_{CH_4}$ and not zero as should be according to the paper.

So suggested general solution does not satisfy both of used boundary conditions.

2. My second big concern is rationality of building this model in its current state. I suppose that each new model should provide substantial improvement of available models. But in your paper only one class of available models is described and improved (models of Curry, Ridgwell, Potter; further CRP models). To my knowledge there are much better models of methane consumption (as example, Saggar et al., 2007; Zhuang et al., 2013). The main their advantage is description of methane consumption and soil methane diffusion not as constant along the soil profile (like in your model and CRP models) but as dependent on soil depth. These models also take into account all environmental controls considered in your paper. So it is not correct to ignore them.

That's why it is necessary:

- to tell that these models exist and to give their brief description
- to explain why it is imporntant to build a new model and why your model is better than others. This explanation is necessary to give comparing your model with CRP models too. You consider the same factors as CRP models, so what are the reasons to improve these models? Are CRV models or models from (Saggar et al., 2007; Zhuang et al., 2013) predict measured methane fluxes worse than your model or not good enough?

3. Using L, the depth of total methane consumption, is good idea, but total methane consumption (or consumption up to 0.1% of atmospheric methane level) does not occur in natural upland soils. There is a certain threshold of methane consumption by microorganisms. Methane consumption stops if this threshold is reached because microorganisms cannot get enough energy by methane oxidation for cell maintenance (Stackhouse et al, 2017). According to literature methane concentration is never close smaller than 0.1 ppm in deep soil horizons and consumption declines to zero in deep soil layers – about 50-70 cm (Bender and Conrad (1992), Whalen et al (1992), Czepiel et al (1995), Priemé and Christensen (1997), Jensen and Olsen (1998)). To my knowledge biological consumption of methane was not ever investigated on a depth more than 1 m in upland soils. Threshold of consumption varies depend on ecosystem type, climate and is defined by oxidation efficiency of methanotrophs (see references above and Stackhouse et al, 2017).

That's why I think that this approach of using total methane consumption depth is not correct.

I think, paper can be published only if all these three problems will be solved.

Minor comments

1. Page 5, rows 15-20. It would be better to give here any estimates, why only diffusive transport and biological consumption should be considered. What about convective transport? Is it always can be omitted? Why term on a right side of eq. (2) is not important? Are conditions really always steady state?

2. Page 10. What is the reason of using old Moldrup paper (same as in Curry paper) while there is much more recent and better soil gas diffusion model in (Moldrup et al., 2013)?

3. Page 30, row 11. Please fix, Sabrekov (like in list of references), not Savrekov.

References

Bender M and Conrad R 1992 Kinetics of CH4 oxidation in oxic soils exposed to ambient air or high CH4 mixing ratios FEMS Microbiology Letters, 101, 261–70

Czepiel P M, Crill P M and Harriss R C 1995 Environmental factors influencing the variability of methane oxidation in temperate zone soils J. Geophys. Res.: Atmos. 100 9359–64

Jensen S and Olsen R A 1998 Atmospheric methane consumption in adjacent arable and forest soil systems Soil Biology and Biochemistry 30 1187–93

Moldrup P., Chamindu Deepagoda T.K.K., Hamamoto S., Komatsu T., Kawamoto K., Rolston D.E., de Jonge L.W. Structure-dependent water-induced linear reduction model for predicting gas diffusivity and tortuosity in repacked and intact soil // Vadose Zone Journal. 2013. V. 12.

Priemé A and Christensen S 1997 Seasonal and spatial variation of methane oxidation in a Danish spruce forest Soil Biology and Biochemistry 29 1165–72

Saggar S, Hedley C B, Giltrap D L and Lambie S M 2007 Measured and modelled estimates of nitrous oxide emission and methane consumption from a sheep-grazed pasture Agriculture, Ecosystems & Environment, 122, 357–65

Stackhouse, B., Lau, M. C. Y., Vishnivetskaya, T., Burton, N., Wang, R., Southworth, A., Whyte, L. and Onstott, T. C. (2017), Atmospheric CH4 oxidation by Arctic permafrost and mineral cryosols as a function of water saturation and temperature. Geobiology, 15: 94–111. doi:10.1111/gbi.12193

Whalen S C, Reeburgh W S and Barber V A 1992 Oxidation of methane in boreal forest soils: a comparison of seven measures Biogeochemistry 16 181–211

Zhuang, Q., Chen, M., Xu, K., Tang, J., Saikawa, E., Lu, Y., ... & McGuire, A. D. (2013). Response of global soil consumption of atmospheric methane to changes in atmospheric climate and nitrogen deposition. Global Biogeochemical Cycles, 27(3), 650-663.

---

## Referee Comment (RC2) · Anonymous Referee #2 · 31 Jul 2017

Comments on "Soil methanotrophy model (MeMo v1.0): a process-based model to quantify global uptake of atmospheric methane by soil" submitted by Fabiola Murguia-Flores et al. to Geoscientific Model Development

General comments

In this manuscript, the authors present a new process-based model of upland soil oxidation by microbes, MeMo. They showed major results on global methane uptake, its latitudinal and spatial distribution, and seasonal change, in comparison with previous models by Potter et al. (1996), Ridgewell et al. (1999), and Curry (2007). I agree that global methane budget is gathering attention in terms of global climate change and so

that the topic is timely.

The manuscript provides a detailed description of basic concept and equations, mathematical solution, and environmental dependencies. I know that GMD accepts such a descriptive paper but still want to recommend shortening main text to some extent. The results presented in this manuscript are basic and lack scientific novelty; again, main text can be truncated by removing redundant statements of results in figures and tables.

I'm not clearly sure what is the substantial advancement of the MeMO model, in comparison with previous models, because the new model used the similar framework for modeling soil methane oxidation. In fact, the estimated global total (34.3 Tg CH4/yr) is around the middle of the previous estimates (Table 7), and one apparent advantage is the better agreement with recent observations. In this regard, the low methane oxidation in humid tropics simulated by MeMO seems reasonable in comparison with previous ones. On the other hand, my serious concern is on the nitrogen limitation factor. The author seems to consider only atmospheric deposition, but in reality, fertilizer and manure input is much more important as nitrogen input into croplands. Previous models, Ridgwell et al. (1999) and Curry (2007), implicitly accounted for the effect by using land-cover data. If this is correct, the MeMo model underestimated the effect of nitrogen input on methane oxidation (as shown in Figure 9).

Overall, I conclude that the manuscript needs major revision and would be reconsidered. I also recommend reinforcing discussion part with respect to implications to experimental and observational studies and potential impacts on climate projections and management.

Specific comments

1. Page 2 Line 20: Please cite more recent syntheses of global methane budget (e.g., Saunois et al., 2016, 2017)

2. Page 8 Figure 1: Please show the atmospheric CH4 concentration for this example.

3. Page 14 Line 1: "Grosso" should be "Del Grosso".

References

Saunois M, Bousquet P, Poulter B, Peregon A, Ciais P, Canadell JG, et al. The global methane budget: 2000–2012. Earth System Science Data 2016, 8: 697–751.

Saunois M, Bousquet P, Poulter B, Peregon A, Ciais P, Canadell JG, et al. Variability and quasi-decadal changes in the methane budget over the period 2000–2012. Atmospheric Chemistry and Physics Discussions 2017: doi:10.5194/acp-2017-5296.

---

## Author Comment (AC1) · 13 Sep 2017

We thank the reviewer for providing these comments and provide responses below. Reviewer comments are in bold followed by our responses.

**Major comment 1: My first big concern is about correspondence between your solution of eq. (2) (using eqs. (3) and (6-7)) and boundary conditions. […] So suggested general solution does not satisfy both of used boundary conditions. eqs. (3) and (6-7)) and boundary conditions. […] So suggested general solution does not satisfy both of used boundary conditions.**

We apologize for an error in the sign of the denominator in Eq. (7) that resulted in this comment. The correct expression for the integration constant B is:

$$B = \frac{c_{CH4} * exp\left(-\sqrt{\frac{k_d}{D_{CH4}}}L\right)}{\left[exp\left(-\sqrt{\frac{k_d}{D_{CH4}}}L\right) - exp\left(\sqrt{\frac{k_d}{D_{CH4}}}L\right)\right]} \quad\quad (7)$$

Inserting A and B into Eq. (3) and solving Eq. (3) for z = 0 and z = L now yields the correct boundary conditions $CH_4$ (0) = $C_{CH4}$ and $CH_4$ (L)=0. Equation 7 has been corrected in the revised manuscript. We thank the reviewer for bringing this error to our attention.

The incorrect equation also was present in the code, which produced L values that were too large. However, because the majority of $CH_4$ is consumed at shallow depth in soil, the overestimation of L resulted in a <1% error in estimates of regional uptake of atmospheric $CH_4$. We have corrected the MeMo code and rerun the simulations. The changes have not significantly altered the modelling outcomes or conclusions of our study and all numbers have been updated throughout the manuscript.

We note also that use of the term "99.9% consumption" has been changed to 'complete consumption' throughout the manuscript.

**2. My second big concern is rationality of building this model in its current state. I suppose that each new model should provide substantial improvement of available models. But in your paper only one class of available models is described and improved (models of Curry, Ridgwell, Potter; further CRP models). To my knowledge there are much better models of methane consumption (as example, Saggar et al., 2007; Zhuang et al., 2013). The main their advantage is description of methane consumption and soil methane diffusion not as constant along the soil profile (like in your model and CRP models) but as dependent on soil depth. These models also take into account all environmental controls considered in your paper. So it is not correct to ignore them. Ridgwell, Potter; further CRP models). To my knowledge there are much better models of methane consumption (as example, Saggar et al., 2007; Zhuang et al., 2013). The main their advantage is description of methane consumption and soil methane diffusion not as constant along the soil profile (like in your model and CRP models) but as dependent on soil depth. These models also take into account all environmental controls considered in your paper. So it is not correct to ignore them.**
**That's why it is necessary:**
- **to tell that these models exist and to give their brief description**
- **to explain why it is important to build a new model and why your model is better than others. This explanation is necessary to give comparing your model with CRP**

**models too. You consider the same factors as CRP models, so what are the reasons to improve these models? Are CRV models or models from (Saggar et al., 2007; Zhuang et al., 2013) predict measured methane fluxes worse than your model or not good enough?**

We developed MeMo to be a process-based global model for simulating past, present and future uptake of atmospheric $CH_4$ by soil. We chose to build on the Potter et al. (1996), Ridgwell et al. (1999) and Curry (2007) (PRC) models because mechanistic simulation of global $CH_4$ consumption in soil could be forced using data from past archives, modern records and future simulations of climate. We acknowledge that higher resolution models and more complex approaches presently exist for modelling soil methanotrophy, in particular, at a local scale; however, comprehensive global datasets that contain driving data at an adequate spatial and temporal resolution are not available (specific examples are discussed below and in text that has been added to the manuscript).

We have not provided detailed descriptions of non-PRC class models in the manuscript because MeMo builds on the PRC models and demonstrating the advances offered by MeMo was the focus our manuscript. The reviewer notes the Zhang *et al.* (2013) global model (hereafter referred to as 'Z13') for simulating soil uptake of atmospheric $CH_4$, which can be regarded as separate from the PRC class of models. The general analytical solution used in Z13 is the same as Curry (2007; C07), which has been improved in MeMo; however, Z13 incorporates differences in its parameterization of microbial activity that are based upon redox potential and maximum rates of $CH_4$ consumption instead of using a base rate for $CH_4$ oxidation. The Z13 model also differs in that it employs modelled ecosystem-specific inputs for Q10 and optimum soil moisture; however, that complexity requires that Z13 operate within the Terrestrial Ecosystem Model (TEM) because global data sets for parameters such as optimum soil moisture and redox potential are not available. In short, driving data for the PRC and MeMo models are only a portion of the input needed for Z13 simulations and consequently, it was not possible to conduct the same level of comparison between MeMo and Z13 that was conducted for the PRC models. However, we note that the Z13 model was not ignored in our original manuscript and that a comparison of global soil uptake of atmospheric $CH_4$ simulated by Z13 and MeMo (and a range of other models) was provided in Table 7. The similarity of the global uptake results is a notable outcome despite differences in the modelling approaches used to simulate $CH_4$ uptake by soil. It is important to note, however, that our study is the first time that a soil methanotrophy model has been validated against global observations, highlighting the importance of accurately quantifying regional variations.

As stated by the reviewer there are biochemical models available at present that are more complex than MeMo (and Z13). These models (e.g., NZ-DNDC and XHAM) have been used to simulate $CH_4$ dynamics at specific sites based upon coupled reaction transport equations that require highly depth-resolved local input data (e.g., Saggar *et al.*, 2007; Oh *et al.*, 2006; Sabrekov *et al.*, 2016). All of these models can be driven by depth-variable parameters when high resolution local data are available; however, the models are impractical for global simulations of soil methanotrophy because of the limited availability of the high resolution global data required to drive the models (e.g. rhizosphere depth, specific soil management, specific metabolic data, enzyme concentrations).

In summary, attributes of MeMo that advance the state of global simulation of soil uptake of atmospheric $CH_4$ are (i) its use of an analytical (more complete) solution to quantify the depth and maximum consumption of atmospheric $CH_4$, (ii) its ability to quantify the influence of

internal CH$_4$ sources (e.g., methane produced in anoxic microsites in soil) on soil methanotrophy and the impact of autochthonous CH$_4$ on regional uptake of atmospheric CH$_4$ by soil linked to seasonal or inter-annual changes in soil moisture or temperature, and (iii) its standalone nature, similar to the PRC models which it is built upon, that eliminates the need to operate within more complex models that provide driving data, and (iv) a detailed validation of simulations both globally and regionally against currently available CH$_4$ uptake rates for soil methanotrophy.

We did not originally describe all available models in the manuscript because MeMo builds explicitly on the PRC class of soil methanotrophy models. However, we concur with the reviewer that the manuscript would be improved by noting how MeMo differs from these other types of models, in particular Z13. We recognize that addition of the new text is contrary to the second reviewer's recommendation that the manuscript be shortened. The following text has been added on page 3 beginning at line 20 (replacing text formerly from line 20 page 3 to line 19 page 4) to address the concerns raised by reviewer 1:

"Several detailed biogeochemical models have been developed to quantify consumption of atmospheric CH$_4$ by soil. Saggar et al. (2007) produced a modified version (NZ-DNDC) of DNDC (Li et al., 2000) to evaluate local impacts of changes in climate, soil properties, fertiliser management and grazing regimes on soil methanotrophy. Sabrekov et al. (2016) developed a process-based model of soil CH$_4$ uptake that also incorporates rhizosphere methanotrophy. Oh et al. (2016) developed a model (XHAM) that explicitly simulates high affinity methanotrophy and active microbial biomass dynamics. These models are driven by high resolution local data sets, which presents challenges for conducting global simulations of soil methanotrophy because of limited availability of input data necessary to drive the models (e.g., global rhizosphere depth, specific soil management, specific metabolic data, enzyme concentrations).

Previous global models included Potter et al. (1996) (hereafter referred to as model 'P96'), which estimates terrestrial uptake of CH$_4$ by calculating diffusive flux of atmospheric CH$_4$ into soil using a modified version of Fick´s first law. Ridgwell et al. (1999) (hereafter referred to as model 'R99') improved the P96 model by explicitly accounting for microbial CH$_4$ oxidation in soil. The R99 model quantifies CH$_4$ oxidation rates as a function of soil temperature, moisture and N content. The latter parameter was estimated using agricultural land area as a proxy for fertilizer application. Solution of the resulting one-dimensional diffusion-reaction equation was approximated semi-numerically assuming steady state conditions. Curry (2007) (hereafter referred to as model 'C07') employed a steady state analytical solution of the one-dimensional diffusion-reaction equation and introduced a scalar modifier to account for the regulation of CH$_4$ oxidation rates by soil moisture and the impact of temperature below 0°C. The C07 model continued to use the R99 agricultural land area approximation to evaluate the effect of N loading on CH$_4$ uptake. The C07 model is employed as a reference model for the Global Carbon Project (Saunois et al., 2016) to estimate global CH$_4$ uptake in dynamic global vegetation models, such as the Lund-Potsdam-Jena model (LPJ-WHy-Me; Wania et al., 2010; Spahni et al., 2011).

The model of Zhang et al. (2013) (hereafter referred to as model 'Z13') employs the same steady state analytical solution as model C07; however, parameterization of microbial activity in model Z13 is based upon redox potential, ecosystem-specific inputs for Q10 and optimum soil moisture, and maximum rates of CH$_4$ consumption instead of a base rate for CH$_4$ oxidation. Consequently, model Z13 operates within the Terrestrial Ecosystem Model (TEM) that provides the necessary driving data because global data sets for many of these parameters are not available. If external data were available, model Z13 presumably could be operated

independently of the TEM in a manner similar to models P96, R99 and C07. However, such a stand-alone application (i.e. decoupled from TEM) would require a new implementation or presumably significant modifications to the code.

We have chosen to focus on refining the R99 and C07 models because availability of new observational and experimental data present an opportunity to re-evaluate global simulations of soil methanotrophy based upon an enhanced version of these models. For example, new global datasets quantifying N deposition and N input via fertilizers now enable better representation of this key inhibitory effect on soil uptake of atmospheric $CH_4$ (Lamarque et al., 2013). In addition, a new global inventory of $CH_4$ uptake rates in soil (Duataur and Verchot, 2007) provides a means to better compare and valid model simulations.

**3. Using L, the depth of total methane consumption, is good idea, but total methane consumption (or consumption up to 0.1% of atmospheric methane level) does not occur in natural upland soils. There is a certain threshold of methane consumption by microorganisms. Methane consumption stops if this threshold is reached because microorganisms cannot get enough energy by methane oxidation for cell maintenance (Stackhouse et al, 2017). According to literature methane concentration is never close smaller than 0.1 ppm in deep soil horizons and consumption declines to zero in deep soil layers – about 50-70 cm (Bender and Conrad (1992), Whalen et al (1992), Czepiel et al (1995), Priemé and Christensen (1997), Jensen and Olsen (1998)). To my knowledge biological consumption of methane was not ever investigated on a depth more than 1 m in upland soils. Threshold of consumption varies depend on ecosystem type, climate and is defined by oxidation efficiency of methanotrophs (see references above and Stackhouse et al, 2017).**

**That's why I think that this approach of using total methane consumption depth is not correct.**

We thank the reviewer for this suggestion. It is straightforward to incorporate a $CH_4$ threshold, $CH_4$ min, in Eqs. 6 and 7 for the case $CH_4$ (L) = $CH_4$ min. In the original manuscript $CH_4$ min = 0 but it is now a variable that can be set in the model:

$$A = -\frac{C_{CH4} * exp\left(\sqrt{\frac{k_d}{D_{CH4}}}L\right) - CH4min}{\left[exp\left(-\sqrt{\frac{k_d}{D_{CH4}}}L\right) - exp\left(\sqrt{\frac{k_d}{D_{CH4}}}L\right)\right]} \tag{6}$$

$$B = \frac{-CH4\,min + C_{CH4} * exp\left(-\sqrt{\frac{k_d}{D_{CH4}}}L\right)}{\left[exp\left(-\sqrt{\frac{k_d}{D_{CH4}}}L\right) - exp\left(\sqrt{\frac{k_d}{D_{CH4}}}L\right)\right]} \tag{7}$$

Eq. (8) becomes:

$$0 = -D_{CH4}\sqrt{\frac{k_d}{D_{CH4}}} \frac{\left(2\,C_{CH4} - CH4min * exp\left(-\sqrt{\frac{k_d}{D_{CH4}}}L\right) - CH4min * exp\left(\sqrt{\frac{k_d}{D_{CH4}}}L\right)\right)}{\left[exp\left(-\sqrt{\frac{k_d}{D_{CH4}}}L\right) - exp\left(\sqrt{\frac{k_d}{D_{CH4}}}L\right)\right]} - F_{CH4} \tag{9}$$

To evaluate the effect of using a $CH_4$ threshold of 0 or 0.1 ppm, we compared the two scenarios. Figure 1 shows that the difference in L between $CH_4$ (L) = 0 and $CH_4$ (L) = 0.1 ppm is ~5 cm, which will have a minimal impact on $CH_4$ uptake flux because the majority of $CH_4$ is consumed

in the top 10 to 30 cm of soil. Based on these changes and analysis we made the following modifications to the manuscript:

1) Updated Eqs. (6), (7) and (9) in the text.
2) Added Figure R1 to the Supplementary file (page 1; Figure S1).
3) Replaced Eq. (9) in the MeMo code for calculation of L.

[Figure]

Figure R1 (also Supplementary Fig S1): Comparison of model-derived depth L when CH₄(L) = 0 (top left), CH₄ (L) = 0.1 ppm (top right), the difference in depth L using the two approaches (bottom left), and CH₄ consumption profiles in soil and total uptake flux using fixed parameters of k, D and CH₄ (bottom right).

Under optimal conditions for methanotrophy, a CH₄ min = 0.1 ppm threshold results in a reduction in L of 6 cm (Supplementary Figure R1 bottom right panel); however, conditions for methanotrophy vary spatially and temporally, and hence use of the 0.1 ppm CH₄ threshold globally yields an average L reduction of 5 cm. The impact on CH₄ uptake rates is negligible because ~90% of atmospheric CH₄ entering soil is consumed within 10 cm of the ground surface. The effect on L size is important when CH₄ min is at least > 0.35 ppm, for example when CH₄ min =1.0 ppm the uptake flux decreases by ~57%.

Thus, the inclusion of a 0.1 ppm threshold for soil methanotrophy does not have an impact on the estimation of the global uptake of atmospheric CH₄, when compared with a scenario in which it is assumed that all CH₄ entering soil is consumed.

**4. Page 5, rows 15-20. It would be better to give here any estimates, why only diffusive transport and biological consumption should be considered. What about convective transport? Is it always can be omitted? Why term on a right side of eq. (2) is not important? Are conditions really always steady state?**

Several studies have shown that soil methanotrophy is limited by $CH_4$ diffusion and that advective fluxes (convective fluxes do not operate at this scale) play only a minor role in $CH_4$ under particular circumstances (Striegl, 1993; Kruse et al., 1996). Regardless, advective fluxes can be readily incorporated in the model (see below) although cannot be parameterized or constrained because of a lack of driving data.

To incorporate advective flux in MeMo an additional advective term is added to the diffusion-reaction equation. Assuming the following boundary conditions: (i) $C(0) = C0$ and (ii) $\frac{dCH_4}{d_z}|_{z->\infty} = 0$ the solution of the advection-reaction-diffusion equation is given by:

$$J_{CH4} = -D_{CH4}(A * a + B * b) - wC$$

Where $w$ is the advective velocity and C is defined as:

$$C(z) = A * \exp(a * z) + B * \exp(b * z)$$

Where:

$$a = \frac{\left(w - \sqrt{w^2 + 4 * D_{CH4} * k_d}\right)}{2 * D_{CH4}}$$

$$b = \frac{\left(w + \sqrt{w^2 + 4 * D_{CH4} * k_d}\right)}{2 * D_{CH4}}$$

Thus, if $w = 0$ the solution is Eq. 10 currently in the manuscript. Solution of the equation using different values of $w$ yields the $CH_4$ depth profiles shown in Figure R2 below.

[Figure]

Figure R2: Calculation of $CH_4$ flux using different values of downward advective velocity ($w$). The depth ($z$) in soil at which $CH_4$ is 0.1 ppm occurs is shown in each panel.

The analysis shows that an advective velocity of 0.01 $cm^2$/s can reduce the depth (L) of complete $CH_4$ consumption by up to 20% under optimal conditions. An advective velocity of 0.1 $cm^2$/s (half the rate of diffusion) can cause a decrease of up to 50%. However, as stated initially no data exist at present to parameterize or valid incorporation of advection into soil methanotrophy models.

**4. Page 5, rows 15-20. Are conditions really always steady state?**

It is reasonable to assume steady state conditions in global models such as MeMo because the timescale of boundary condition changes is long compared to the time required to attain steady state conditions in soil.

**2. Page 10. What is the reason of using old Moldrup paper (same as in Curry paper) while there is much more recent and better soil gas diffusion model in (Moldrup et al., 2013)?**

We have cited the Moldrup et al. (2013) paper in our work. While the authors evaluate several soil-diffusion models, it is important to note that their performance was only slightly better than the one employed in MeMo:

$$\frac{Dp}{Do} = \Phi^{4/3} \left(\frac{\Phi_{air}}{\Phi}\right)^{1.5+3/b}$$

The new version proposed by Moldrup et al. (2013) is:

$$\frac{Dp}{Do} = \Phi_{air}{}^2 \left(\frac{\Phi_{air}}{\Phi}\right)$$

Where $Dp$ is the gas diffusion coefficient in soil ($cm^3$ air $cm^{-1}$ soil $s^{-1}$), $Do$ is the gas diffusion coefficient in free air ($cm^2$ air $s^{-1}$), $\Phi$ is total pore volume ($cm^3$ $cm^{-3}$), $\Phi_{air}$ is air-filled porosity ($cm^3$ $cm^{-3}$), $b$ is a scalar that accounts for soil structure ($b = 15.9\, f_{clay} + 2.91$).

The new version of the gas diffusion equation from Moldrup et al. (2013) provides only a marginal improvement in the RSME fit (0.017; Figure 2 in Moldrup et al., 2013) versus the model we used in MeMo (RSME=0.028). However, the main reason that we use this formulation of the equation is that the new model no longer includes the soil structure parameter (b) that accounts for the effects of clay content on gas diffusion in soil. Data for this parameter are available globally for different soil types which enables a more explicit assessment of the impact of soil texture on global uptake of atmospheric CH4.

**3. Page 30, row 11. Please fix, Sabrekov (like in list of references), not Savrekov.**

This error has been corrected.

**References**

Moldrup, P., Iversen, N.: Modeling Diffusion and reaction in soils: II Atmospheric Methane Diffusion and consumption in a forest soil, Soil Sci. 161, 355-365, 1996.

Kruse, C.W., Moldrup, P., Iversen, N.: Modeling Diffusion and reaction in soils: II Atmospheric Methane Diffusion and consumption in a forest soil, Soil Sci. 161, 355-365, 1996.Stange, F., Butterbach-Bahl, K., Papen, H.: A process-oriented model of N2O and NO emissions from forest soils: 1. Model development, J. Geophys. Res. Atmospheres 105, 4369–4384, doi:10.1029/1999JD900949, 2000.

Li, C., Aber, J., Stange, F., Butterbach-Bahl, K., Papen, H.: A process-oriented model of N2O and NO emissions from forest soils: 1. Model development, J. Geophys. Res. Atmospheres 105, 4369–4384, doi:10.1029/1999JD900949, 2000.Trugman, A.T., Moch, J., Onstott, T.C., Jørgensen, C.J., D'Imperio, L., Elberling, B., Emmerton, C.A., St. Louis, V.L.,

Medvigy, D.: A scalable model for methane consumption in arctic mineral soils, Geophys. Res. Lett. 43, 2016GL069049, doi:10.1002/2016GL069049, 2016.

Oh, Y., Stackhouse, B., Lau, M.C.Y., Xu, X., Trugman, A.T., Moch, J., Onstott, T.C., Jørgensen, C.J., D'Imperio, L., Elberling, B., Emmerton, C.A., St. Louis, V.L., Medvigy, D.: A scalable model for methane consumption in arctic mineral soils, Geophys. Res. Lett. 43, 2016GL069049, doi:10.1002/2016GL069049, 2016., A.F., Glagolev, M.V., Alekseychik, P.K., Smolentsev, B.A., Terentieva, I.E., Krivenok, L.A., Maksyutov, S.S.: A process-based model of methane consumption by upland soils. Environ, Res. Lett., 11, 075001, doi:10.1088/1748-9326/11/7/075001, 2016.

Sabrekov, A.F., Glagolev, M.V., Alekseychik, P.K., Smolentsev, B.A., Terentieva, I.E., Krivenok, L.A., Maksyutov, S.S.: A process-based model of methane consumption by upland soils. Environ, Res. Lett., 11, 075001, doi:10.1088/1748-9326/11/7/075001, 2016.Giltrap, D.L., Lambie, S.M.: Measured and modelled estimates of nitrous oxide emission and methane consumption from a sheep-grazed pasture, Agric. Ecosyst. Environ, 122, 357–365, doi:10.1016/j.agee.2007.02.006, 2007.

Saggar, S., Hedley, C.B., Giltrap, D.L., Lambie, S.M.: Measured and modelled estimates of nitrous oxide emission and methane consumption from a sheep-grazed pasture, Agric. Ecosyst. Environ, 122, 357–365, doi:10.1016/j.agee.2007.02.006, 2007.Wania, R., Neef, L., van Weele, M., Pison, I., Bousquet, P., Frankenberg, C., Foster, P.N., Joos, F., Prentice, I.C., van Velthoven, P.: Constraining global methane emissions and uptake by ecosystems, Biogeosciences 8, 1643–1665, doi:10.5194/bg-8-1643-2011, 2011.

Spahni, R., Wania, R., Neef, L., van Weele, M., Pison, I., Bousquet, P., Frankenberg, C., Foster, P.N., Joos, F., Prentice, I.C., van Velthoven, P.: Constraining global methane emissions and uptake by ecosystems, Biogeosciences 8, 1643–1665, doi:10.5194/bg-8-1643-2011, 2011., R., Ross, I., Prentice, I.C.: Implementation and evaluation of a new methane model within a dynamic global vegetation model: LPJ-WHyMe v1.3.1., Geosci Model Dev 3, 565–584, doi:10.5194/gmd-3-565-2010, 2010.

Wania, R., Ross, I., Prentice, I.C.: Implementation and evaluation of a new methane model within a dynamic global vegetation model: LPJ-WHyMe v1.3.1., Geosci Model Dev 3, 565–584, doi:10.5194/gmd-3-565-2010, 2010.Saikawa, E., Lu, Y., Melillo, J.M., Prinn, R.G., McGuire, A.D.: Response of global soil consumption of atmospheric methane to changes in atmospheric climate and nitrogen deposition, Glob. Biogeochem. Cycles 27, 650–663, doi:10.1002/gbc.20057, 2013.

Zhuang, Q., Chen, M., Xu, K., Tang, J., Saikawa, E., Lu, Y., Melillo, J.M., Prinn, R.G., McGuire, A.D.: Response of global soil consumption of atmospheric methane to changes in atmospheric climate and nitrogen deposition, Glob. Biogeochem. Cycles 27, 650–663, doi:10.1002/gbc.20057, 2013.

---

## Author Comment (AC2) · 13 Sep 2017

Reply submitted as supplement

Please also note the supplement to this comment:
https://www.geosci-model-dev-discuss.net/gmd-2017-124/gmd-2017-124-AC2-supplement.pdf

---

## Referee Report (RR1)

Review of "Soil Methanotrophy Model (MeMo v1.0): a process-based model to quantify global uptake of atmospheric methane by soil", Report №2.

Major comments

1. Page 4, line 6. First of all, please, fix the reference, it should be ZhUang, not Zhang. Second, I think, sentence "The model of Zhang et al. (2013) (hereafter referred to as model 'Z13') employs the same steady state analytical solution as model C07" is incorrect. Z13 uses steady state reaction-diffusion equation for methane (as almost all discussed models), but this equation was solved *numerically* for entire soil depth from 0 m to 1 m (theoretically it is not possible to solve this equation analytically in this case). Thus Z13 takes into account vertical heterogeneity of methane consumption controls, which is not the case for models of Potter family and your model also. It is principle advantage of Z13 and other recent models in comparison with your model. It definitely should be mentioned in a paper text.

I also do not understand why you write in the same paragraph "However, such a stand-alone application (i.e., decoupled from TEM) would require a new implementation or presumably significant modifications to the code." It is not a disadvantage of the Z13 model, it is a technical issue.

Zhuang, Q., Chen, M., Xu, K., Tang, J., Saikawa, E., Lu, Y., ... & McGuire, A. D. (2013). Response of global soil consumption of atmospheric methane to changes in atmospheric climate and nitrogen deposition. Global Biogeochemical Cycles, 27(3), 650-663.

2. This is not good that you do not check presentation of your model against MEASURED methane fluxes. I see that you use for field data for parametrization how temperature, moisture and nitrogen influence on methane consumption. And you illustrate it with nice figures. But you do not show model presentation against independent flux data set. It is important because influence of methane consumption controls is often not independent from each other and multicollinearity does exist in this case. So it can be dangerous to use model to the global flux calculations and predictions without a validation using independent methane flux data.

It is not always the problem. For example, for Z13 authors also did not do this validation. But they use numerous flux data sets to obtain values of different parameters in optimization procedure. In this way Z13 takes into account interaction between controls, although this algorithm is very often mathematically incorrect (it is so called ill-posed inverse problem).

We can see MeMo presentation for different latitudes against combined methane flux data in comparison with other models of Potter family (Fig. 5), and MeMo seems to give substantial improvement. But in data set from (Dutaur and Verchot, 2007) almost all flux measurements are not seasonal average but measurements made in several days or weeks during season. You also do not use data obtained in sites where fluxes from (Dutaur and Verchot, 2007) data set were measured. In summary, I think that there is a lack of validation in your model because in Fig. 5:

- you compare modeled seasonal fluxes and sporadically measured fluxes

- you did not validate your model for set of sites with their own ecological parameters; instead you compare fluxes measured in multiple sites of 10° regions with modeled latitude average seasonal flux.

That is why instead of pointwise convergence necessary for predictions you showed only convergence in general, when model accuracy for certain geographical points and sites is hidden.

I suggest to mention that you did not validate your model directly and to explain why you did not do it. At least for sites used for validation in Z13 it should be possible.

Minor comments

1. I recommend to remove Figures 1 and 2. They do not give any deep insights in the subject. I think ideas presented in these Figures are obvious without visualization for almost all readers of GMD. You can write several sentences instead.

2. If you use the same model for soil gas diffusivity as previous models I recommend to shorten section about it (2.3.1).

Summary

In my opinion, the manuscript was significantly improved after first round of corrections. Introduction now give much better representation of the state of the art in methane consumption modeling and the scope of current paper. Mistakes in math were fixed, some results of the model became much more realistic (for example – estimates of L). Authors made a big efforts to upgrade paper text, I really appreciate this.

But I think, some aspects still need to be fixed. Moreover, I think that there is a lack of model validation.

---

## Referee Report (RR2)

**General Comments**

Murguia-Flores et al., presented a process-based CH4 consumption model to quantify global soil CH4 consumption. This version has appropriately addressed comments from two previous reviewers including their major concerns. The topic is timely and appropriate for GMD. And, the model development, parameterization, global extrapolation, and inter-model comparison are all written and convincingly presented. Below are some of my specific but minor comments.

**Specific comments**

P1L16 potent greenhouse gas

P1L22 at the global scale

P1L26 "We show that the improved representation of these key drivers of soil methanotrophy results in a better fit to observational data." Actually, it's hard to tell is the better model-data fit coming from process representation, driver representation, or just parameterization. But it's totally fine to conclude that the model improved structurally and parametrically.

P2L5 preindustrial era

P3L11 interannual variability and uncertainty

P5table 1. Values for some critical constants are missing (e.g., kd, A, B)

P13Table 3.  What's the uncertainty of MeMo k0 parameters?

P19 Figure 4. It's actually a little bit ambiguous that rN is parameterized with N input rate. With the same N input rate (gNm2y-1), one can fertilize the system with a monthly frequency verses a daily frequency. Then the actual N retained in the soil will be totally different across the year. Thus, the same N input rate could have different inhibition controls on CH4 consumption.

P21 Table 7 global soil CH4 uptake has mean value and uncertainty. It's not clear in the manuscript, where the uncertainty is from? In particular, why the uncertainty is so large in observation but the uncertainty is so small in MeMo.

---

## Author Response (AR2)

**Referee # 1**

We thank the referee for the comprehensive and detailed comments, which have helped to improve the manuscript. Please find our responses to the referee's comments (indicated in bold) below.

**1. Page 4, line 6. First of all, please, fix the reference, it should be ZhUang, not Zhang.**

Thank you for highlighting this mistake. The text has been changed accordingly.

2. I think, sentence "The model of Zhang et al. (2013) (hereafter referred to as model 'Z13') employs the same steady state analytical solution as model C07" is incorrect. Z13 uses steady state reaction-diffusion equation for methane (as almost all discussed models), but this equation was solved *numerically* for entire soil depth from 0 m to 1 m (theoretically it is not possible to solve this equation analytically in this case). Thus Z13 takes into account vertical heterogeneity of methane consumption controls, which is not the case for models of Potter family and your model also. It is principle advantage of Z13 and other recent models in comparison with your model. It definitely should be mentioned in a paper text.

Thank you for the observation. We agree that the main difference between Z13 and the Potter family of models (including MeMo) is how the reaction-transport equations are solved. MeMo solves the equation analytically while Zhuang solves it numerically using multiple soil layers. A numerical approach allows for resolving the vertical heterogeneity of, for instance, diffusion coefficients, soil moisture and, thus, yields better results for local simulations and/or when soil properties are well-characterized or when comprehensive ecosystem model output (e.g. TEM) is available. Yet, MeMo has been designed with the aim to investigate past and future dynamics of the global methane soil sink over large timescales (e.g.  $10^2-10^5$  years). These applications require a computational efficient solution, as well as the ability to constrain model parameters based on (paleo)climate model outputs. We therefore argue that the numerical approach does not necessarily represent an improvement over MeMo for such computationally expensive and data limited applications.

We now include a sentence explaining this substantial difference between the models:

P4L6 "The model of Zhuang et al. (2013) (hereafter referred to as model 'Z13') employs the same steady state reaction-diffusion equation for CH4 as previous models; however, Z13 solves the steady state reaction-diffusion equation for CH4 numerically using multiple soil layers"

3. I also do not understand why you write in the same paragraph "However, such a stand-alone application (i.e., decoupled from TEM) would require a new implementation or presumably significant modifications to the code." It is not a disadvantage of the Z13 model, it is a technical issue.

While it is a technical issue, it is one that makes it prohibitive to easily implement the Z13 scheme. To decouple the Z13 model from TEM would require a significant modification of the code and possibly the model because there is no realistic way to include multiple soil layers without TEM. We feel our text correctly captures the technical challenge of adapting Z13 to other applications.

4. This is not good that you do not check presentation of your model against MEASURED methane fluxes. I see that you use for field data for parametrization how temperature, moisture and nitrogen influence on methane consumption. And you illustrate it with nice figures. But you do not show model presentation against independent flux data set. It is important because influence of methane consumption controls is often not independent from each other and multicollinearity does exist in this case. So it can be dangerous to use model to the global flux calculations and predictions without a validation using independent methane flux data.

We disagree with this comment and are puzzled as the reviewer mentions the comparison against flux data in the comment below. No previous global model of soil uptake has ever been validated against a global database of observations and we, for the first time, conduct a regional-scale validation (Figure 5 and Section 4.2 describe this regional validation against external observations).

A point-by-point comparison to data from Dutuar and Verchot (2007) is not appropriate for a global model. To do so would require comparison of site specific data with the closest location in a coarsely resolved model grid from MeMo. The comparison would not be meaningful because global models such as MeMo are designed to represent regional scale dynamics and not fine scale conditions. Consequently, we have validated our global model through comparison to regional ecosystem scale data, which is standard practice. Notably, our study is the first to date to perform such a global validation of a soil methanotrophy model. We show in Figure 5 how MeMo performs in a regional-scale validation compared to previous models, demonstrating that soil uptake of CH4 in the tropics is greatly over-estimated in other models.

5.- We can see MeMo presentation for different latitudes against combined methane flux data in comparison with other models of Potter family (Fig. 5), and MeMo seems to give substantial improvement. But in data set from (Dutaur and Verchot, 2007) almost all flux measurements are not seasonal average, but measurements made in several days or weeks during season. You also do not use data obtained in sites where fluxes from (Dutaur and Verchot, 2007) data set were measured. In summary, I think that there is a lack of validation in your model because in Fig. 5:

- you compare modeled seasonal fluxes and sporadically measured fluxes

- you did not validate your model for set of sites with their own ecological parameters; instead you compare fluxes measured in multiple sites of 10° regions with modeled latitude average seasonal flux.

That is why instead of pointwise convergence necessary for predictions you showed only convergence in general, when model accuracy for certain geographical points and sites is hidden.

I suggest to mention that you did not validate your model directly and to explain why you did not do it. At least for sites used for validation in Z13 it should be possible.

The comparison between MeMo and field observations from Dutaur and Verchot (2007) was performed because the latter are the best data available on which to base a comparison. Dutaur and Verchot (2007) use local estimates to upscale to annual flux at each location and to total global uptake.

We acknowledge the limitations of performing a validation of a coarsely resolved global model against site observational data (as described in the previous comment); however, our approach represents the first step towards understanding model limitations. This simple validation demonstrates that previous models greatly overestimate soil uptake of atmospheric methane across the tropics because their parameterizations of soil moisture and k0 were not correct.

We now state in the text on Page 22 Line 14, "The latitudinal distribution of soil uptake rates of atmospheric CH4 predicted using the R99 and C07 models, and MeMo are shown in Fig. 5 accompanied by direct measurements of CH4 oxidation rates from Dutaur and Verchot (2007) and a 10° running average. We chose to validate MeMo and previous models against regionally averaged observations to conduct the comparison at scales resolved by global models such as MeMo. This model is not intended to represent fine-scale site-specific attributes of soil but rather broad regional soil characteristics and CH4 uptake fluxes."

**5. I recommend to remove Figures 1 and 2. They do not give any deep insights in the subject. I think ideas presented in these Figures are obvious without visualization for almost all readers of GMD. You can write several sentences instead.**

Figures 1 and 2 are important to include to show the setup of the model. Figure 1 illustrates the computational solution of L, which is a fundamental difference between MeMo and previous models. Figure 2 is important because it shows the sensitivity of CH4 uptake to k0 and thus the importance of the parameter. This detail is fundamental because k0 differs greatly from previous models.

**If you use the same model for soil gas diffusivity as previous models I recommend to shorten section about it (2.3.1).**

Thank you for this suggestion; however, for clarity we prefer to include all equations in the text with appropriate citations rather than redirecting readers to other publications for that information.

We thank Referee #2 for the positive assessment of this work and for providing comprehensive reviews. Please find our response to the referee's comment (indicated in bold) below.

**Page 34 Line 32: for CRU data, now we can use TS 3.25 or TS 4.0.1 from web site. (This is not mandatory but optional).**

Thank you for drawing our attention to the availability of these new data sets. For future applications, we will use updated forcing data, as they become available.

We thank the referee for the detailed comments. Referee comments are shown in bold, followed by our response.

**P1 L16 potent greenhouse gas**

The text has been changed to 'potent greenhouse gas.'

**P1 L22 at the global scale**

The text has been changed to 'at the global scale.'

P1 L26 "We show that the improved representation of these key drivers of soil methanotrophy results in a better fit to observational data." Actually, it's hard to tell is the better model-data fit coming from process representation, driver representation, or just parameterization. But it's totally fine to conclude that the model improved structurally and parametrically.

Thank you for this observation. We have changed the text to indicate that the improved model-fit comes from both the structure and parameterization of MeMo.

P1 L26 "We show that the improved structural and parametric representation of key drivers of soil methonotrophy in MeMo results in a better fit to observational data."

**P2 L5 preindustrial era**

The text has been changed to 'since the pre-industrial era' on Page 2 Line 16

**P3 L11 interannual variability and uncertainty**

The text has been changed to 'large inter-annual variability and uncertainty.'

**P5 table 1. Values for some critical constants are missing (e.g., kd, A, B)**

Thank you for this observation; however, as far as we can tell all base variables and constants are included in the table (along with their values). The ones mentioned in the reviewer's comment are constructed from other variables and constants, and therefore were not included in the table. For example, kd is a variable that depends on soil moisture, soil temperature, k0 and nitrogen as explained in section 2.3.2. A and B are integration constants, whose values depend on L, kd, and  $D_{CH4}$  as defined by Eqs. 6 and 7. Integration constants A and B have now been removed from Table 1 to be consistent with our statement that only base variables and constants are presented in the table.

**P13 Table 3. What's the uncertainty of MeMo k0 parameters?**

This is an interesting point. Both C07 and R99 did not report the uncertainty of this parameter because of a lack of field measurements. In the case of MeMo, we face the same problem. We discuss the importance of k0 and the sensitivity of CH4 uptake to this parameter in Figure 2 and discuss the scarcity of k0 measurements in the 'Model Limitations' section. We now include a sentence to explaining why k0 uncertainty could not be characterized:

P13 L22 "; however, the uncertainty of this value could not be characterized due to a dearth of available observational data."

P19 Figure 4. It's actually a little bit ambiguous that rN is parameterized with N input rate. With the same N input rate (gNm2y-1), one can fertilize the system with a monthly frequency verses a daily frequency. Then the actual N retained in the soil will be totally different across the year. Thus, the same N input rate could have different inhibition controls on CH4 consumption.

Thank you for this interesting comment. We agree that nitrogen fertilization will have temporal structure that is not resolved by MeMo. For this reason, we used long-term field observations to parameterize rN in addition to data from laboratory experiments. Both approaches yield a similar pattern. We have modified the figure legend to ensure it is clear that field measurements were obtained from long-term data.

P19 Figure 4 caption: "CH4 uptake response as a function of nitrogen deposition and fertilizer application factor  $r_N$ . The linear fit (black line) is based on observations from field (long-term) and laboratory measurements (gray and blue dots; Supplementary 1, Table S3)."

**P21 Table 7 global soil CH4 uptake has mean value and uncertainty. It's not clear in the manuscript, where the uncertainty is from? In particular, why the uncertainty is so large in observation, but the uncertainty is so small in MeMo.**

The uncertainty in MeMo corresponds to temporal variability over the time period of the study (1990-2009). The large uncertainty in the observations results from: (i) the lack of representation of several ecosystems, and (ii) measurements being conducted during specific seasons rather than a full annual cycle. In the global observational dataset, Dutaur and Verchot (2007) standardized the data to annual values and across global ecosystems, thus providing global estimates. Nonetheless, these limitations result in a large uncertainty and we have added a paragraph explaining how the uncertainty was calculated in MeMo and in the observations.

P21L6 "MeMo predicts an average annual global flux of  $33.5 \pm 0.6$  Tg CH4 y-1 for the period 1990 to 2009. Uncertainty in this flux was calculated as the standard deviation of annual global CH4 uptake."

P21L12 "Upscaling of field measurements of soil methanotrophy rates from 120 different studies spanning a wide range of ecosystems yielded an uptake flux of  $36 \pm 23$  Tg CH4 y-1 (Dutaur and Verchot, 2007). The large uncertainty associated with the mean flux results from differences in data representation for ecosystems and a tendency for sampling to be conducted seasonally rather than annually."